# Striated muscle-specific base editing enables correction of mutations causing dilated cardiomyopathy

Markus Grosch [1,2,3], Laura Schraft [1], Adrian Chan[4], Leonie Küchenhoff [1], Kleopatra Rapti[5], Anne-Maud Ferreira[2], Julia Kornienko [1,3], Shengdi Li[1], Michael H. Radke [6,7], Chiara Krämer[5], Sandra Clauder-Münster[1], Emerald Perlas[8], Johannes Backs[3,9], Michael Gotthardt [6,7,10], Christoph Dieterich [3,4], Maarten M. G. van den Hoogenhof [3,9], Dirk Grimm [3,5,11] & Lars M. Steinmetz [1,2,3,12] ✉

Dilated cardiomyopathy is the second most common cause for heart failure with no cure except a high-risk heart transplantation. Approximately 30% of patients harbor heritable mutations which are amenable to CRISPR-based gene therapy. However, challenges related to delivery of the editing complex and off-target concerns hamper the broad applicability of CRISPR agents in the heart. We employ a combination of the viral vector AAVMYO with superior targeting specificity of heart muscle tissue and CRISPR base editors to repair patient mutations in the cardiac splice factor *Rbm20*, which cause aggressive dilated cardiomyopathy. Using optimized conditions, we repair >70% of cardiomyocytes in two *Rbm20* knock-in mouse models that we have generated to serve as an in vivo platform of our editing strategy. Treatment of juvenile mice restores the localization defect of RBM20 in 75% of cells and splicing of RBM20 targets including TTN. Three months after injection, cardiac dilation and ejection fraction reach wild-type levels. Single-nuclei RNA sequencing uncovers restoration of the transcriptional profile across all major cardiac cell types and whole-genome sequencing reveals no evidence for aberrant off-target editing. Our study highlights the potential of base editors combined with AAVMYO to achieve gene repair for treatment of hereditary cardiac diseases.

Next-generation CRISPR tools enable gene repair of disease-associated mutations in situ, in the organ of interest, thereby achieving complete prevention or cure of the disease[1]. To date, a few clinical trials have been initiated applying CRISPR in vivo to treat mutations causing blindness, high cholesterol, or protein aggregation[2,3]. Hundreds of pathogenic single-nucleotide variants (SNVs) have been associated with cardiac diseases, making the heart an attractive target for gene therapy[4]. However, few attempts have been made to correct heritable cardiac disorders in vivo. Several studies have corrected pathogenic cardiac mutations in mouse[5–7] and human embryos[8], which has major ethical considerations, and also requires prior knowledge of the inherited mutation. Others have disrupted exons by Cas9-mediated non-homologous end joining in mice[9], dogs[10] and pigs[11] entailing the danger for erroneous DNA repair, which could impair gene expression.

Cardiomyocytes are non-proliferating cells that are impervious to homology-directed gene repair, the method of choice for installing

precise genome edits. Recently, CRISPR base editors have been developed to allow efficient nucleotide conversions in vivo in post-mitotic cells[1]. Thus, we evaluated the use of base editors for the treatment of familial dilated cardiomyopathy (DCM), a severe form of heart disease and the second most common cause of heart failure[12]. Treatment options for DCM patients include drugs to reduce blood pressure or block the neurohormonal system. However, with a 15-year survival rate of only 34%[13], the mortality of DCM patients receiving this treatment is very high[14]. We focused on mutations in *RBM20*, found in 3% of patients with aggressive, early onset DCM[15]. Patients with familial *RBM20*-DCM normally harbor a single-nucleotide disease-causing variant[16] making it a prime target for base editors to install single-nucleotide conversions. *RBM20* encodes a cardiac splice factor that regulates alternative splicing of genes critical for the function of cardiomyocytes[16]. *RBM20* mutations are enriched in a small stretch of six amino acids within the RS-domain and were recently shown to result in aberrant formation of cytoplasmic granules, which likely amplify the disease phenotype[17–19].

Besides correcting the mutation, the major goal for any CRISPR-related gene therapy is to attain organ-specific gene delivery to reduce the chance of potentially deleterious off-target editing. Due to their low risk of immunogenicity[20] and integration[21], as well as their high amenability to genetic retargeting to desired organs, adeno-associated viruses (AAVs) present one of the safest and most versatile options for gene delivery. Previous cardiac gene transfer has been performed with the serotype AAV9 despite its predominant targeting of the liver upon intravenous injection[22]. We have recently identified a synthetic variant of AAV9, named AAVMYO, which exhibits high target affinity for muscle cells including cardiomyocytes and low affinity for other organs such as the liver[22]. Here, we leverage AAVMYO for systemic delivery of base editors to cardiomyocytes, the main cell type expressing *Rbm20*. We optimize the strategy to selectively repair two pathogenic mutations in *Rbm20*'s RS-domain resulting in near-complete prevention of the disease phenotype in mice, with no evidence for guide RNA (gRNA)-dependent off-target activity.

## Results

### P635L and R636Q *Rbm20* knock-in mice exhibit a DCM phenotype

Adenine base editors (ABEs) convert adenines (A) to guanines (G) and have been used successfully in emerging clinical trials[23]. Since none of the existing *Rbm20* animal models are amenable to ABE-mediated nucleotide conversion, we generated two mouse models harboring G > A mutations. Specifically, we established two *Rbm20* knock-in mouse models with the amino acid substitutions P635L and R636Q, respectively, orthologous to the *RBM20* mutations P633L and R634Q in humans previously identified in DCM patients (Supplementary Fig. 1a)[24]. No significant changes were observed in *Rbm20* mRNA expression in these mice (Supplementary Fig. 1b). We performed deep phenotyping to identify aberrant molecular signatures and physiological traits that could be rescued upon base editing. We focused on RBM20 localization, gene expression and heart function since these parameters are dysregulated in mice, humans and pigs with RBM20 RS-domain mutations[17–19].

Immunostaining of isolated cardiomyocytes showed that homozygous (HOM) P635L and R636Q mutant mice have cytoplasmic RBM20 granules, indicating mislocalization of the mutant RBM20 protein from its normal nuclear localization (Fig. 1a). The heterozygous (HET) mutants diverged strongly in the degree of RBM20 mislocalization. While RBM20 was predominantly nuclear in P635L HET, it formed small cytoplasmic granules in R636Q HET mice (Fig. 1a–c). RNA-sequencing (RNA-seq) revealed that the number of differentially expressed genes (DEGs) compared to wild-type (WT) was sixfold higher in R636Q HET compared to P635L HET but lower than in P635L and R636Q HOM mice (Fig. 1d and Supplementary Data 1). DEGs common for both P635L and R636Q exhibited dose-dependency between HET and HOM (Supplementary Fig. 2). Gene ontology (GO) analysis of the common DEGs revealed dysregulation of genes involved in muscle function and metabolic genes (Fig. 1e). The expression of natriuretic peptide precursors A and B (*Nppa* and *Nppb*), which are biomarkers of heart failure[25], was substantially elevated in HOM mice (Supplementary Fig. 1c). We identified 58 differentially spliced genes (DSGs) common for both P635L and R636Q HOM mice (Supplementary Data 2) with the predominant splice event being exon skipping (Fig. 1f). These DSGs were associated with muscle and cytoskeletal functions (Supplementary Fig. 1d). Notably, approximately half of all DEGs and DSGs did not overlap between P635L and R636Q HOM (Supplementary Fig. 1e). While these specific genes could suggest the presence of mutation-specific downstream processes, their *P* values were higher on average than for overlapping genes (Supplementary Fig. 1f). This is consistent with the detection of subtle changes in transcript abundance arising due to biological variation, such as between mice, or other confounding factors that were detected by our deep RNA-seq with 100 Mio. reads on average per genotype. While no difference in the abundance of splice events between P635L and R636Q HET was observed (Fig. 1f), a subset of crucial RBM20 targets including *Ttn*, *Camk2d* and *Tpm2* were more dysregulated in R636Q HET compared to P635L HET (Fig. 1g). We performed RT-PCR and qPCR to validate the isoforms of *Ttn*, *Camk2d*, *Ryr2* and *Ldb3*, which were differentially expressed in mutant mice, and observed stronger dysregulation of *Ttn* and *Ldb3* in R636Q HET compared to P635L HET mice (Fig. 1h and Supplementary Fig. 1g). P635L and R636Q HOM exhibited similar levels of aberrant splicing and stronger than the HET mice (Fig. 1g, h and Supplementary Fig. 1g).

Next, we investigated whether these molecular differences in both mouse models affected the cardiac phenotype. Survival curves indicated that both P635L and R636Q HOM mice died prematurely in the first 120 days albeit to a lesser extent than other *Rbm20* RS-domain mutations (survival rate: 78% P635L, 81% R636Q, 66% S637A[26], 51% S639G[27]) (Fig. 1i). Notably, we backcrossed our mutant mice to C57BL/6J where others have used C57BL/6N[28], which could explain the differences in the survival as C57BL/6N is more susceptible to cardiac deterioration upon pressure overload[29]. Both histological analysis and gene expression did not uncover major signs of fibrosis in 16-week-old mutant mice except upregulation of the fibrosis marker *Col1a2* and *Mmp2* in R636Q HOM mice (Supplementary Fig. 1h–j). The clinical definition of DCM is based on an ejection fraction of <45% and left ventricular dilation[12]. We performed narcosis echocardiography, which confirmed that both mouse models exhibit a DCM phenotype with significantly reduced ejection fraction (Fig. 1j). However, they displayed only minor increase in cardiac volume (except in P635L HOM) and no significant change in the left ventricular internal diameter (LVID) (Supplementary Fig. 1k, l). Corroborating the RNA-seq results, and correlating with cytoplasmic granule formation, the ejection fraction was more reduced in R636Q HET compared to P635L HET mice. After 1 year, no significant worsening of the DCM-associated phenotype was observed in ejection fraction and cardiac volume for P635L HET and HOM mice whereas LVID and cardiac volume significantly increased in R636Q HET and HOM mice (Supplementary Fig. 1m–p). All mutant mice exhibited consistently higher LVID and cardiac volume compared to WT. We conclude that P635L and R636Q *Rbm20* mutant mice exhibit DCM characteristics found in animals and patients with other RS-domain mutations[17–19]. For subsequent rescue strategies, we focused on P635L and R636Q HOM mice since they showed a more pronounced molecular and physiological defect enabling better quantification of the efficacy of the base editor treatment.

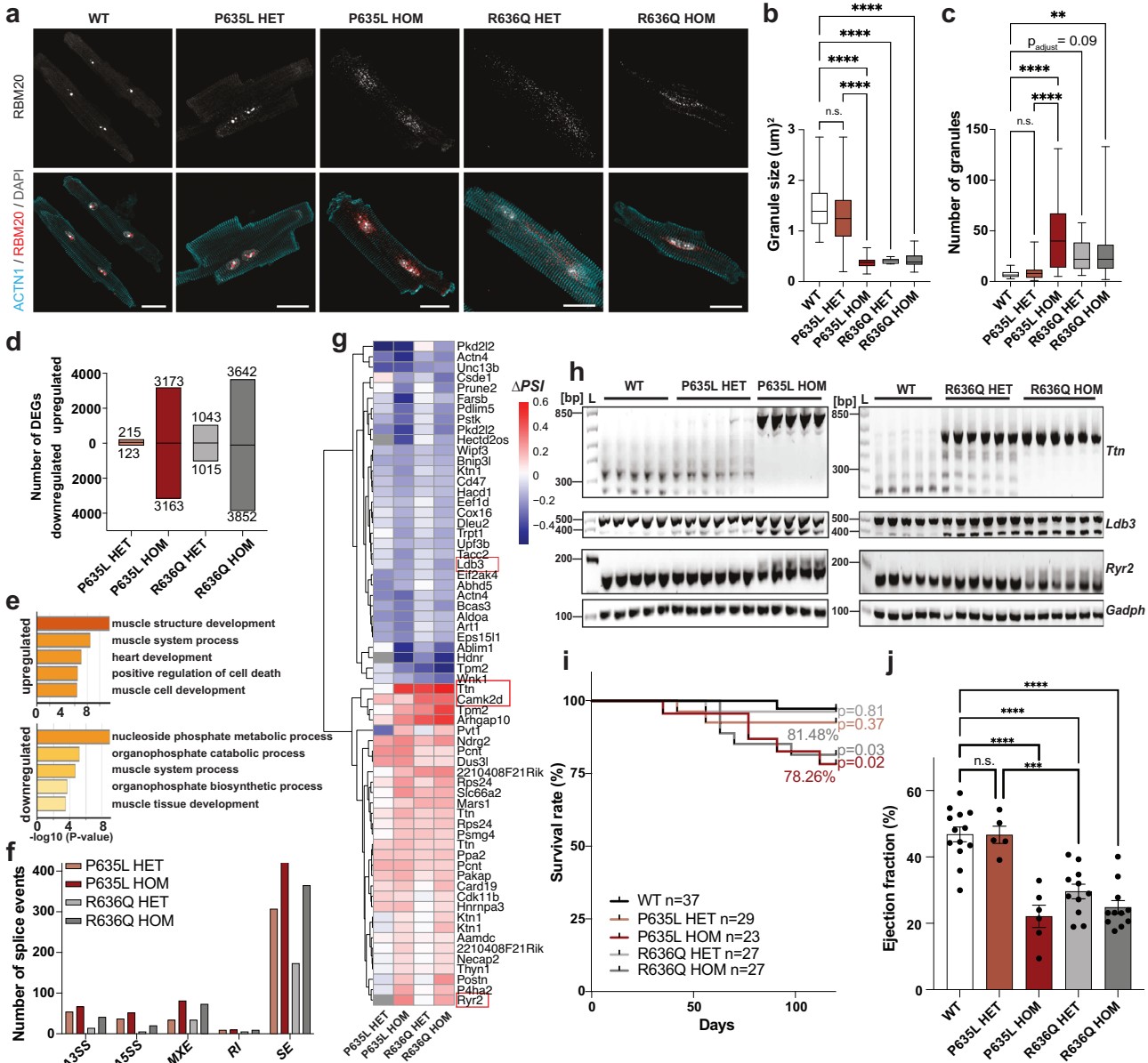

**Fig. 1 | Molecular and physiological characterization of P635L and R636Q mouse lines. a** Confocal images of isolated adult murine cardiomyocytes. Scale bar: 20 μm. ACTN1 was used as cardiomyocyte marker. **b, c** RBM20 granule size (**b**) and amount (**c**) in adult mouse cardiomyocytes. $N = 21$ (WT), 16 (P635L HET), 28 (P635L HOM), 16 (R636Q HET) and 39 (R636Q HOM) images with 1–4 cells each obtained from three mice per genotype. Boxplots depict the median with the box including the 25–75th percentile and the whiskers ranging from the smallest to the largest value. **d** Number of DEGs ($P_{adjust} < 0.05$) in bulk RNA-seq of *Rbm20* mutant mice compared to WT. $N = 5$ mice per genotype. **e** GO analysis (biological function) of DEGs overlapping for both P635L and R636Q HOM mice with a stringent cut-off of $P_{adjust} < 1e^{-10}$ to reduce the number of DEGs for display in Supplementary Fig. 2. **f** Number of differentially spliced events compared to WT detected and categorized by rMATS: alternative 5′ or 3′ splice site (A5SS or A3SS), mutually exclusive exons (MXE), retained intron (RI), skipped exon (SE). **g** Averaged ΔPSI ( = percent spliced-in) values relative to WT of significant differentially spliced events

($P_{adjust} < 0.01$, ΔPSI > 0.1) overlapping in both HOM *Rbm20* mutant mice. Multiple splice events per gene are depicted if they match the selection cut-off. Genes in red were validated by RT-PCR or qPCR. Grey squares indicate that the splice event was not detected by rMATS. **h** RT-PCR of RBM20 target genes *Ttn*, *Ryr2*, *Ldb3* and the housekeeping gene *Gapdh*. **i** Kaplan–Meier survival curve of mutant mice monitored for 120 days. *P* value obtained by Log-rank test between each mutant and WT indicated next to the curves. Percentage of survival indicated for HOM mice. **j** Percentage of ejection fraction determined by narcosis echocardiography of mutant mice. $N = 13$ (WT), 5 (P635L HET), 6 (P635L HOM), 11 (R636Q HET) and 11 (R636Q HOM) mice. *P* values in (**b, c, j**) obtained from one-way ANOVA with Tukey's multiple comparison test: ****$P < 0.0001$, ***$P < 0.001$, **$P < 0.01$, n.s. = not significant. All data were obtained in 16-week-old mice except in (**j**) where data of 24-week-old mice is shown. Error bars depict the standard error of the mean (SEM) in all panels.

## Base editors repair pathogenic *Rbm20* mutations in vitro and in mice

To test the feasibility of base editor treatment for repairing pathogenic P635L and R636Q mutations, we first transfected ABEs combined with compatible gRNAs in proliferating human iPSCs and non-proliferating cardiomyocytes derived from induced pluripotent stem cells (iPSC-CMs) with the orthologous *RBM20* mutations P633L and R634Q (Fig. 2a). Due to sequence restrictions, we used ABEs containing Cas9 that recognize non-canonical PAMs such as "NRN" used in conjunction with the ABE SpRY[30], or the ABEs NRTH / NRCH named after their PAM preference[31]. Moreover, we tested circular permuted ABE (CP-1041) exhibiting a broader editing window[32] for targeting of

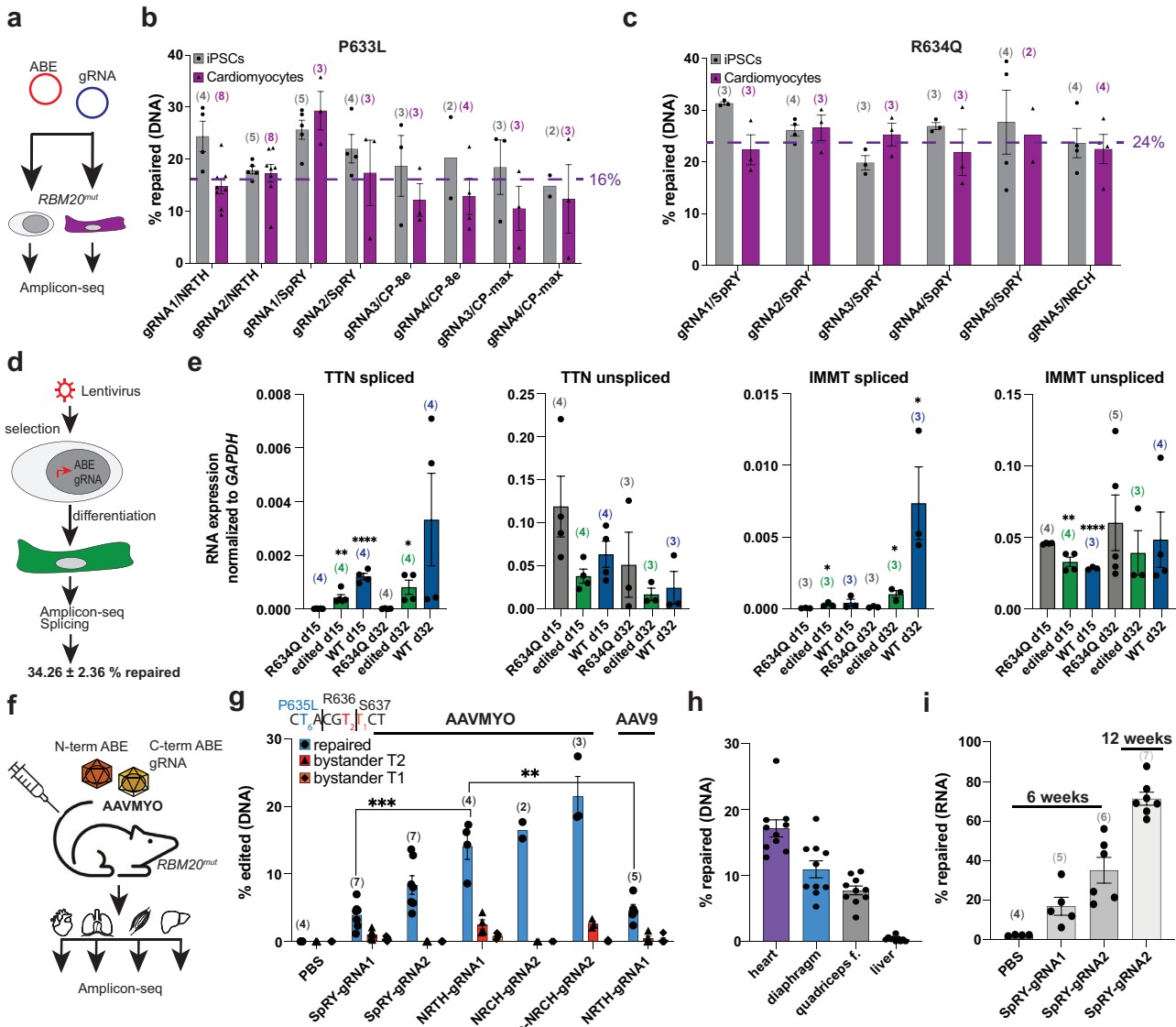

**Fig. 2 | Base editing of *RBM20* in human iPSC-CMs and in mice. a–c** Transient expression of base editor and gRNA in human iPSCs and iPSC-CMs. Experimental outline in (**a**), editing efficiency of P633L in (**b**) and R634Q in (**c**). "CP" labels the circular permuted base editor CP-1041. Purple line indicates average repair efficiency in iPSC-CMs. **d** Generation of stable base editor expression in R634Q iPSCs with a repair efficiency of 34.26 ± 2.36% as determined by amplicon-seq. $N = 3$ independent differentiations. **e** Expression of spliced and unspliced isoforms of *TTN* and *IMMT* in parental, R634Q and edited R634Q iPSC-CMs differentiated for 15 and 32 days. Significant changes compared to R634Q indicated when present and analyzed by unpaired, two-tailed *t* tests. *$P < 0.05$, **$P < 0.01$, ****$P < 0.0001$. **f** Experimental outline of AAVMYO-mediated base editing in mice. **g** Percentage of editing of P635L HOM mice injected with AAVMYO carrying different gRNA-base editor combinations or PBS as empty control. For NRCH-gRNA1, AAV9 was also used as vector. Significance was assessed using unpaired, two-tailed *t* tests

***$P < 0.001$, **$P < 0.01$, *$P < 0.05$. Sequence shows location of the on-target edit in blue and the two bystander edits in red. Numbers depict the position of the nucleotides within the targeting gRNA (gRNA2 was used as reference) with the PAM sequence in position 21–23. **h** Allele frequency of repaired DNA in the muscle tissues heart, diaphragm and quadriceps femoris (quadriceps f.), as well as the liver, plotted for ten mice with the highest editing events in (**g**). **i** Percentage of editing of *Rbm20* mRNA in mice treated with AAVMYO-SpRY for 6 or 12 weeks. Editing was assessed by amplicon-seq of cDNA isolated from the whole heart. Most base editors contain the deaminase variant Abemax except when indicated by "8e", which are base editors with the Abe8e version. Percentage "repaired" in (**b, c, g–i**) is defined by NGS reads from amplicon-seq with only the wild-type sequence. The number of biological replicates i.e., independent differentiations in (**b, c, e**) or mice in (**g, i**) is indicated in brackets above the bars. Error bars depict the SEM in all panels.

canonical PAMs in P633L. We observed comparable editing efficiencies of *RBM20* mutations between iPSCs and iPSC-CMs of up to 30% on average (Fig. 2b, c). No base editor clearly outperformed others. Indel formation, a byproduct of base editors[30,31], was below 2.5% with no significant bias between different ABEs (Supplementary Fig. 3a). Likewise, bystander edits (i.e., unwanted A > G conversions within the gRNA window) were generally below 1% with no significant trend between different base editors except for circular permuted editors, which led to more bystander edits for P633L likely due to their broader editing window (Supplementary Fig. 3b, c). Next, we analyzed whether

editing of iPSC-CMs restored RBM20-mediated splicing. We generated R634Q iPSCs with stable expression of the base editor SpRY together with a targeting gRNA using lentiviral transduction leading to a repair efficiency of 34% (Fig. 2d). After differentiation to iPSC-CMs, the expression levels of spliced isoforms of *TTN* and *IMMT*, prominent RNA targets of RBM20[33], were increased in expression whereas the unspliced isoforms were decreased in base-edited cells suggesting that base editing restored RBM20-related splice defects (Fig. 2e).

Encouraged by these results, we tested the performance of base editors in editing of the heart in vivo. We utilized a split-intein

strategy[34] to package both parts of the ABE (controlled by the constitutive CAG promoter) together with a gRNA expression cassette into the synthetic AAV9 variant AAVMYO (referred to as AAVMYO-ABE) (Fig. 2f). To determine optimal virus concentration for systemic delivery, we used a YFP reporter transgene and observed that 1e12 vector genomes (vg) (corresponds to 8.33e13 vg/kg total virus concentration) ensures high viral targeting of the heart without overt expression of the transgene in the liver (Supplementary Fig. 3d, e). To identify the optimal base editor-gRNA combination, we tested the in vivo editing performance of the ABEs NRTH, NRCH and SpRY. We performed tail vein injections of AAVMYO-ABEs combined with two different gRNAs in P635L HOM mice and analyzed editing of the mutation in the heart, diaphragm, quadriceps and liver after 6 weeks. Experiments were performed in juvenile 4-week-old mice resembling young DCM patients with the possibility to prevent disease progression. We found clear performance differences between both tested gRNAs. gRNA2 displayed higher on-target editing efficiency than gRNA1, and fewer bystander edits (Fig. 2g). The base editors NRCH and NRTH outperformed SpRY with regards to editing efficiency (Fig. 2g), in contrast to in vitro editing where no clear differences between different ABEs were observed. We also tested NRCH conjugated with the latest and most efficient version of adenine deaminase, namely Abe8e[35] (referred to as 8e-NRCH), and observed the highest editing with 21.4% on average (Fig. 2g). This editor, however, also showed bystander edits of 2.7%, the most common bystander edit of which ($T_2$; 2.64%) introduces a synonymous codon change and is likely inconsequential. Of note, due to different positioning of the base editor, a second non-synonymous bystander edit ($T_1$) was observed for gRNA1 in up to 1.31% of reads leading to a codon change from TCT (serine) to CCT (proline). Therefore, the use of gRNA1 was discontinued for subsequent long-term editing and phenotyping. $T_1$ was also detected in 8e-NRCH combined with gRNA1 but only in 0.09% of reads on average. No indels were observed in any condition. For NRTH, we also generated AAV9 vectors, which exhibited less than half of the editing efficacy of the AAVMYO-ABE counterpart supporting the superiority of AAVMYO for cardiac gene delivery (Fig. 2g). Notably, no significant editing was observed in the liver. Highest editing occurred in the heart followed by diaphragm and quadriceps suggesting that the liver and likely other non-muscle tissue are protected from on-target but also off-target base editing activity (Fig. 2h). Viral DNA copy number and relative RNA expression correlated broadly with editing efficiency (Supplementary Fig. 3f). Since AAVMYO predominantly infects cardiomyocytes, which constitute only 30–50% of all cardiac cells[36,37], the viral expression measured in the heart is likely an underestimate.

We also generated an ABE version driven by the promoter of human cardiac troponin T (*hTNNT2*), which led to highly specific editing in the heart and absence of editing in other tissues (Supplementary Fig. 3g). However, *hTNNT2*-driven ABEs were only comparable in editing efficacy with constitutive CAG promoter-driven ABEs when doubling the virus concentration. Since we sought to exclude potential side effects from high viral loads in the mouse model, we continued mainly with CAG-driven ABEs. To evaluate the potential of long-term base-editing, we collected the heart 12 weeks instead of 6 weeks after injection. Whereas 8e-NRCH outperformed SpRY after 6 weeks, both exhibited similar levels of editing after 12 weeks (Supplementary Fig. 3h). Editing of the liver did not exceed 2% even 12 weeks after injection (Supplementary Fig. 3h).

Finally, we quantified the extent of *Rbm20* mRNA that was edited, since this allows estimating the editing efficacy in cardiomyocytes, i.e., the cell type that predominantly expresses *Rbm20* (Supplementary Fig. 3i)[37]. Sequencing of heart cDNA after 6 weeks of editing revealed that on average 35% of mRNA molecules were edited with SpRY compared to 8% on the DNA level (Fig. 2g, i). Strikingly, 12 weeks after editing, 71% of *Rbm20* mRNA were edited on average compared to 18% of DNA (Fig. 2g, i). Similar to the copy number measurements, the

discrepancy between the extent of DNA and RNA editing is likely due to the fact that AAVMYO only infects cardiomyocytes, which overall represent a smaller fraction of the DNA extracted from the heart. We conclude that base editors delivered with AAVMYO enable highly efficient muscle-specific repair of *Rbm20* mutations in mice.

## Base editors repair *Rbm20*-DCM phenotypes in mice

To measure long-term effects, we performed tail vein injections of AAVMYO-ABE, using our best performing editor-gRNA combinations 8e-NRCH and SpRY, the latter exhibiting less bystander edits. These injections were performed in 4-week-old P635L and R636Q HOM mice. Long-term base-editing and physiological effects were analyzed 12 weeks after injection. Amplicon-seq of whole heart gDNA revealed an average repair efficiency of 18–20% in the heart and below 2% in the liver across the two tested *Rbm20* mutations (Supplementary Fig. 4a). As observed before with gRNA2, bystander edits were detected in mice treated with 8e-NRCH but not in mice injected with SpRY (Supplementary Fig. 4b). Compared to editing after 6 weeks, overall, more bystander edits were detected in P635L mice treated with 8e-NRCH. The main synonymous bystander edit $T_2$ occurred in 4.09% of reads on average followed by a missense mutation $T_1$ (0.33%). Two other synonymous mutations $T_{-2}$ (0.20%) and $T_{17}$ (0.42%) were observed (Supplementary Fig. 4b). R636Q mice treated with 8e-NRCH exhibited one bystander edit $A_2$ (0.69%). Notably, we observed bystander edits only in reads that have also received the correct edit indicating that only repaired alleles were prone to bystander edits, which effectively lowers the editing efficacy (Supplementary Fig. 4c). Indels surrounding the gRNA window were not detectable. Also, levels of *Rbm20* mRNA editing were substantially higher, namely 68% for SpRY and >85% for 8e-NRCH (Fig. 3a).

To evaluate the extent of phenotype rescue upon base editing, we performed RBM20 localization and gene expression assays, as well as analysis of heart pump function. RBM20 immunostaining in heart tissue sections revealed eradication of the cytoplasmic RBM20 granules and restoration of the characteristic nuclear RBM20 foci in 75% of cells in AAVMYO-ABE-treated mice compared to control mice injected with saline (Fig. 3b, c). Next, we analyzed splicing of *Ttn* and observed increased expression of the spliced, as well as decreased expression of the unspliced isoforms. Moreover, the splicing profile of other RBM20 targets *Camk2d*, *Ldb3* and *Ryr2* approached levels of the WT control (Fig. 3d and Supplementary Fig. 4d). Since *Ttn* mis-splicing likely contributes to aberrant cardiomyocyte function in RBM20-DCM[38], we validated TTN expression at the protein level. AAVMYO-ABE-treated mice showed reduced expression of the gigantic TTN isoform (G-TTN) from 83 to 17%, with levels of constitutive N2A and N2BA isoforms approaching levels of WT (Fig. 3e, f and Supplementary Fig. 4e). RNA-seq of PBS or ABE-treated P635L and R636Q HOM mice revealed that about 50% of the mis-spliced exons in PBS-treated mice were rescued after base editing; especially *Ttn* exons were amongst the most strongly reverted splice events (Fig. 3g).

Finally, we performed narcosis echocardiography 8 and 12 weeks after injection. After 8 weeks, there was a clear but not significant trend toward an increase in the ejection fraction (Supplementary Fig. 4f). However, after 12 weeks, the ejection fraction was reverted almost to WT levels (Fig. 3h). In line with the restoration of cardiac function, LVID and cardiac volume decreased upon base editing albeit without reaching statistical significance (Fig. 3i, j). Moreover, expression of the heart failure biomarkers *Nppa* and *Nppb* was reduced after base editing compared to PBS-injected samples (Supplementary Fig. 4g). Notably, we also performed echocardiography after injection of hTNNT2-driven ABE and observed significant improvement of the ejection fraction after 12 weeks (Supplementary Fig. 4h). We conclude that AAVMYO-ABE delivery significantly improves the molecular and physiological defects associated with *Rbm20* mutations in mice.

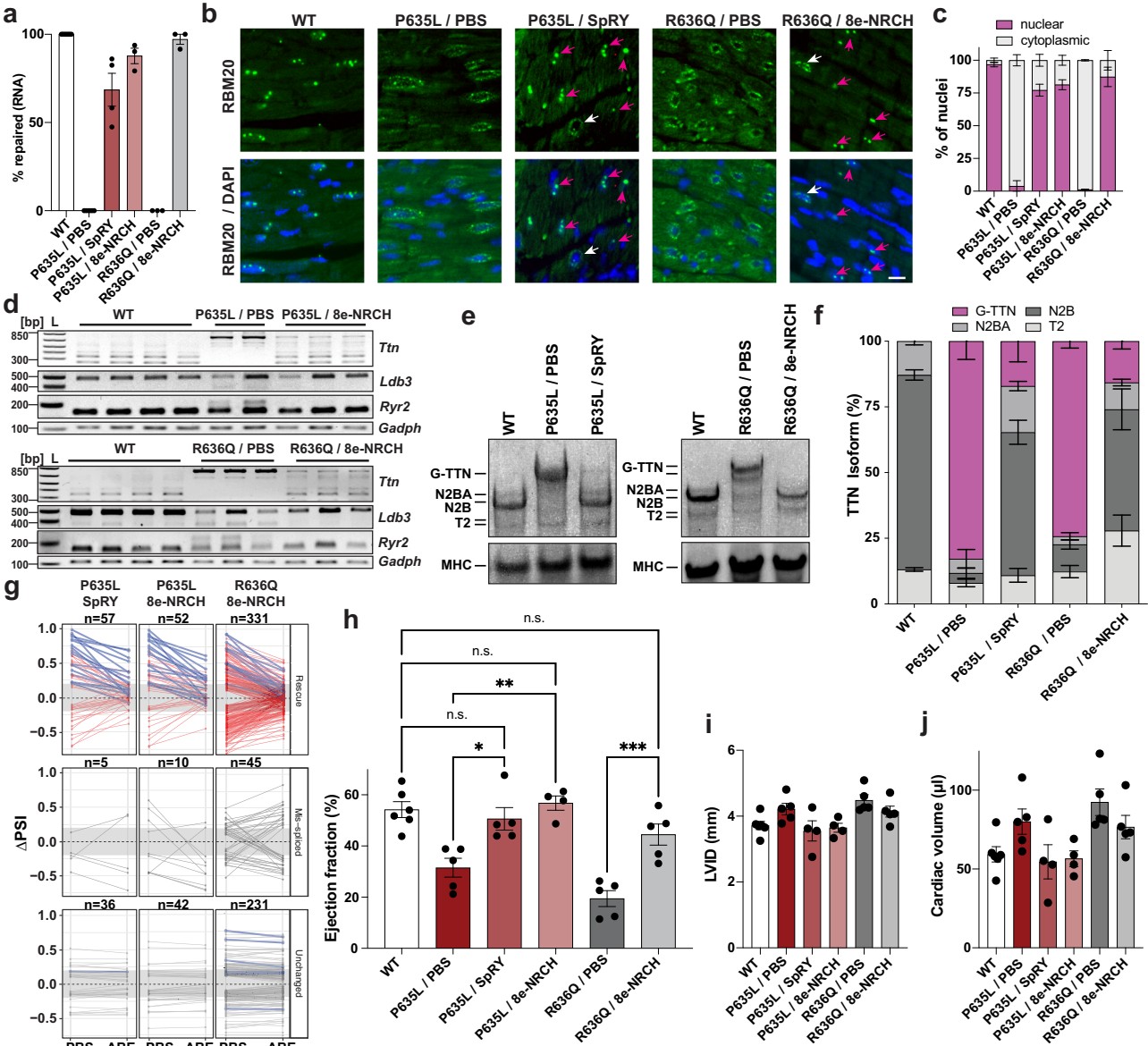

**Fig. 3 | Phenotypic characterization of mice after AAVMYO-ABE treatment.**
**a** Allele frequency of repaired *Rbm20* mRNA in mice treated with AAVMYO-ABE determined by RNA-seq. $N = 3$ (R636Q), 4 (P635L) and 8 (WT) mice per condition. **b, c** RBM20 staining in whole heart tissue sections in WT and *Rbm20* mutant mice treated with PBS or AAVMYO-ABE. Representative images in (**b**) and quantification of nuclear and cytoplasmic RBM20 localization in (**c**). Scale bar: 20 µm. Arrows highlight nuclear restored RBM20 (magenta) and cytoplasmic RBM20 (white) in base-edited mice. Manual quantification of >200 nuclei in 2 mice per condition. **d** Isoform expression of RBM20 target genes *Ttn, Ryr2, Ldb3* and the housekeeping gene *Gapdh* determined by RT-PCR. $N = 2$–4 mice per condition. **e, f** Vertical agarose gel (**e**) and quantification (**f**) of titin protein isoforms in WT and *Rbm20* mutant mice treated with PBS or AAVMYO-ABE. Based on gel images in Supplementary Fig. 4e. $N = 3$ mice per condition except for WT and P635L/SpRY where four mice were analyzed. **g** RNA-seq data showing changes of the ΔPSI values

relative to WT in P635L, or R636Q HOM mice injected with saline or base editor. See 'Methods' section (bulk RNA sequencing and analysis) for definition of the three categories. Rescue splice events are labeled in red, all *Ttn* splice events in blue. $N$ = number of splice events per category. R636Q was sequenced deeper compared to P635L explaining the difference in number of DSGs detected. $N = 3$ mice per condition except for nWT (4 mice) and P635L SpRY (5 mice). **h**–**j** Percentage of ejection fraction (**h**), LVID (**i**) and cardiac volume (**j**) determined by narcosis echocardiography of mutant mice treated with PBS or AAVMYO-ABE. $N = 5$ mice per condition. Same WT cohort used as in Supplementary Fig. 1m–o (16-week time point). $P$ values obtained from one-way ANOVA with Tukey's multiple comparison test: ***$P < 0.001$, **$P < 0.01$, *$P < 0.05$, n.s. = not significant. All data were obtained 12 weeks after AAVMYO-ABE injection. Only P635L or R636Q HOM mice were treated. Error bars depict the SEM in all panels.

## Rescue of cell type-specific gene expression upon base editing

To investigate whether base editing restores the transcriptional landscape of the heart, we performed single-nuclei RNA sequencing (snRNA-seq) of 40,235 nuclei isolated from hearts of 16-week-old mice in the absence and with base editor treatment. We analyzed nuclei from WT ($n = 7867$), P635L HOM ($n = 16,218$), and P635L HOM mouse hearts 12 weeks after injection of AAVMYO with NRCH ($n = 6246$), 8e-NRCH ($n = 2286$) or SpRY ($n = 7618$) (Supplementary

Fig. 5a–c). UMAP projection based on transcriptional similarity and clustering identified 11 major cell types that express known cell type markers found in previous studies[37] (Fig. 4a, b). Sub-clustering within the ventricular cardiomyocytes revealed that cells from base-edited mice have transcriptional profiles between those of WT and P635L HOM mice (Fig. 4c). The fraction of immune cells (lymphoid and myeloid) increased only slightly upon AAVMYO treatment (2.4–4% in WT and P635L HOM, 3.6–5.6% in base-edited mice) indicating the

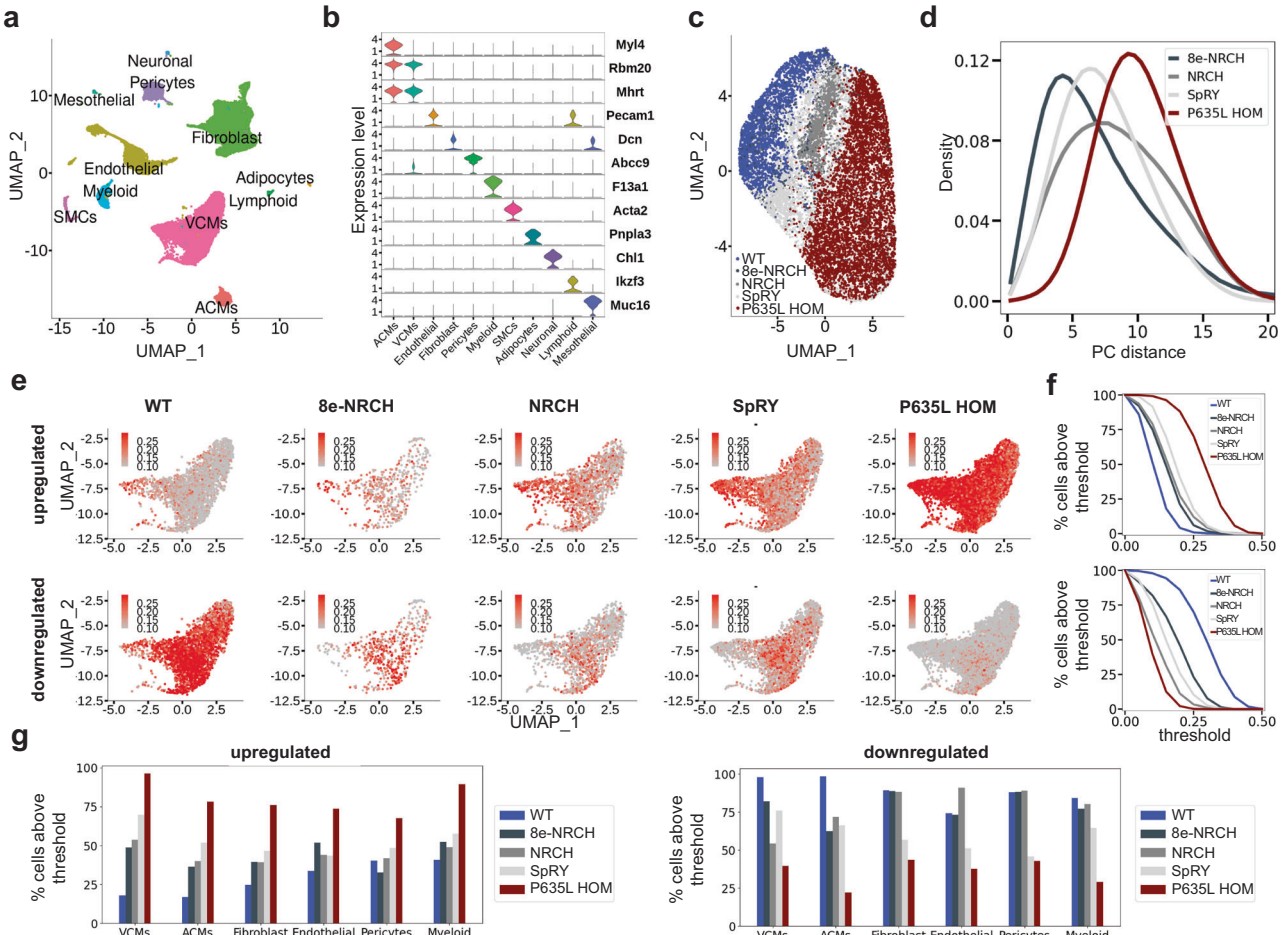

**Fig. 4 | Cell type-specific profiling of cells after base editing by snRNA-seq.**
**a** UMAP projection integrating all datasets and annotated based on their gene expression profile. **b** Expression of known marker genes defining the main cell types. **c** UMAP projection of ventricular cardiomyocytes from WT, P635L HOM, and base-edited mice. **d** Histograms depicting the distribution of pairwise Euclidean distances of ventricular cardiomyocytes from P635L HOM and base-edited mice relative to WT using the two largest principal components (PC). **e** UMAP projection showing the activity score (see 'Methods' section snRNA-seq analysis) of cardiomyocytes using a subset of genes that were up- or downregulated in P635L HOM relative to WT. Maximum 15 significantly up- or downregulated were used. DEGs are listed in Supplementary Data 3. **f** Threshold of activity score values based on (**e**) relative to percentage of cells above the threshold for genes upregulated (upper panel) or downregulated (lower panel) in P635L HOM cardiomyocytes relative to WT. **g** Percentage of cells above the critical threshold for genes upregulated or downregulated in P635L HOM cells relative to WT. VCMs ventricular cardiomyocytes, ACMs atrial cardiomyocytes, SMCs smooth muscle cells. Data were generated by snRNA-seq of isolated nuclei from two mice per condition.

absence of an overt immune response (Supplementary Fig. 5d). In our snRNA-seq data, we also analyzed the expression of the base editor complex itself and confirmed predominant targeting of cardiomyocytes by AAVMYO (Supplementary Fig. 5e, f). Next, we compared transcriptome similarities between WT, P635L HOM and P635L HOM after base editor treatment. In ventricular cardiomyocytes, cells after base editor treatment shifted closer to WT in their transcriptional profile whereas no overt trend was observed for the other major cell types (Fig. 4d and Supplementary Fig. 5g). Since transcriptome effects could be masked by genes unrelated to the *Rbm20* mutation, we analyzed the transcriptomic profile for genes that were significantly dysregulated in P635L HOM mice (based on snRNA-seq, see 'Methods' section snRNA-seq analysis). Ventricular cardiomyocytes from base-edited mice exhibited a gene expression profile that is between WT and P635L HOM, indicating that gene expression was at least partially restored (Fig. 4e–g). Strikingly, we also observed major gene expression changes in other cell types with levels reaching WT levels for atrial cardiomyocytes, pericytes, endothelial cells, myeloid cells and fibroblasts (Fig. 4g and Supplementary Fig. 5h). This indicates that downstream effects associated with RBM20-DCM such as changing the gene expression profile of non-cardiomyocytes were

repaired even though the AAVMYO-ABE treatment specifically targets cardiomyocytes.

## No evidence for AAVMYO-ABE-induced off-target editing
Finally, we sought to identify off-target mutations induced by the base editor. We performed whole-genome sequencing (WGS) in three P635L HOM mice treated with AAVMYO and the base editor SpRY for 12 weeks. For each mouse, we sequenced tail (harvested before the injection), liver and heart tissue with an average genome coverage of 47× (Supplementary Data 4). WGS confirmed a high viral load in the heart, low levels in the liver and background signal in the tail (Supplementary Fig. 6a), with an average on-target allele editing frequency of 27% in the heart and absence of editing in the other tissues (Supplementary Fig. 6b). We adapted a previous strategy[39] (see 'Methods' section whole-genome sequencing and analysis) to identify novel variants for each tissue by overlapping three variant callers that identify SNVs and indels (Supplementary Fig. 6c, d). We focused on variants that were detected by at least two variant callers. After applying additional filter steps, we found on average 208–650 tissue-specific variants in the heart, liver, tail (Fig. 5a). The relative contribution of A > G/T > C nucleotide conversions was not increased in the heart

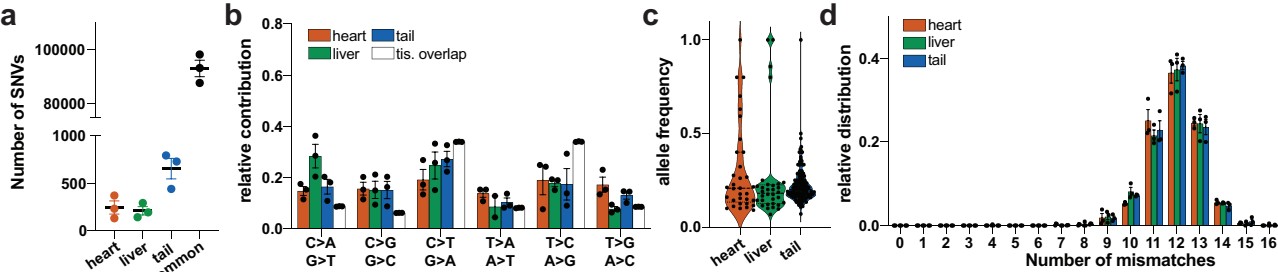

**Fig. 5 | WGS of mouse tissue before and after AAVMYO-ABE treatment. a** Mean number of tissue-specific and variants detected in all tissues (common). **b** Mean relative distribution of distinct nucleotide conversions for tissue-specific and common SNVs. Tissue overlap represent variants that were common to all three tissues. **c** Allele frequency of tissue-specific T > C/A > G variants. *N* = 33 (heart), 32 (liver), 146 (tail). **d** Mean number of mismatches to the gRNA and PAM sequence in the area of ± 30 bases around the variant start site. Error bars depict the SEM in all panels. *N* = 3 mice in (**a**, **b**, **d**).

compared to other tissue-specific variants or variants that overlap in all three tissues (Fig. 5b). Moreover, the allele frequency of A > G/T > C mutations was similar in all three tissues (Fig. 5c), indicating the absence of systematic off-target mutations installed by this ABE. The genomic distribution of tissue-specific variants was similar to common variants with only a small fraction of SNVs in exonic regions (Supplementary Fig. 6e). None of the heart-specific variants were shared between the three replicates and seven were identified in two mice. Only one SNV was found to change the amino acid sequence of a coding gene (Supplementary Data 5). No sequence homology was detected in the genomic region surrounding the novel variants compared to the gRNA sequence, which suggests the absence of gRNA-dependent editing (Fig. 5d). We further analyzed 16 selected sites by amplicon-seq: 7 loci with highest sequence similarity to the gRNA used and 9 candidate A/T > G/C variants determined by WGS with the highest sequencing coverage (Supplementary Fig. 6f). No SNVs were detected in the in silico predicted off-target sites. In addition, 4 out of 9 candidate loci from WGS were >90% mutated in both PBS and AAVMYO-ABE-treated mice and therefore are likely germline variants. For the remaining 5 SNVs, no difference in the percentage of editing was observed after AAVMYO-ABE injection compared to PBS. Overall, this data does not indicate the presence of ABE-induced DNA off-target edits.

Since RNA editing has been reported as byproducts of ABEs[40,41], we also analyzed bulk RNA-seq data obtained 12 weeks after AAVMYO-ABE treatment of P635L and R636Q HOM mice. We confirmed high expression of the base editor in the heart and its absence in the liver (Supplementary Fig. 7a) leading to high on-target and minor bystander editing for 8e-NRCH in P635L HOM (Supplementary Fig. 7b). Unbiased variant detection was performed on the RNA-Seq data (Supplementary Fig. 7c; details in the method) and a small but significant increase (from 17% to 19%) in the fraction of A > G mutations was observed only in the 8e-NRCH compared to PBS-treated R636Q HOM mice (Supplementary Fig. 7d). No significant differences were observed in the other AAVMYO-ABE-treated samples. Similar to WGS, we could not detect major sequence homology between the region surrounding the variants and the gRNA (Supplementary Fig. 7e) indicating that the increased frequency of A > G mutations is not due gRNA-dependent effects. In summary, while our analysis prohibits the detection of random SNVs arising in a subset of cells, we do not find evidence that the base editing strategy induces systemic off-target editing.

## Discussion

Chances of success of any gene repair strategy depend on the degree of editing reached in the target cell type. This efficiency is contingent on the extent of editor delivery in the target cell type and its gene editing efficacy within each cell. Since *Rbm20* is mainly expressed in cardiomyocytes, we could infer overall editing efficiency by analyzing *Rbm20* mRNA. Strikingly, 70–87% of cardiomyocytes were repaired,

demonstrating both the superior targeting capability of AAVMYO and the efficacy of base editors, specifically ABEs. In a recent study, Nishiyama and colleagues employed base editors delivered by AAV9 in 5-day-old mice and achieved an *Rbm20* mRNA editing efficiency of 66% on average[28]. However, with a total concentration of 2.5e14 vg/kg, 3-times more virus was used than in our study where we treated larger, juvenile 4-week old mice. The differences in AAV amount can be attributed to promoter choice and AAV serotype. Nishiyama et al. employed a TNNT2 (cTnT) promoter to confer cardiac-specific base editor expression, which in our hands required twice the AAV amount to reach editing efficiencies similar to expression controlled by the constitutive CAG promoter. Using a similar strategy but combined with thoracic injections, another study recently achieved 81% editing efficiency of cDNA in the left ventricle[42]. Notably, since AAVMYO selectively targets cardiomyocytes and not the liver, less virus is required for cardiac delivery compared to AAV9. In our hands, base editors delivered by AAV9 were only half as efficient in correcting *Rbm20* mutations compared to AAVMYO-ABE when injected in the same concentrations. The advantages of this higher targeting specificity are two-fold. First, high AAV concentrations are associated with toxicity due to AAV-induced adaptive immune response[43]. Second, AAV production is a time-consuming and costly process. Especially for human therapies, up to 1.5e17 vg of virus have been applied[44] and using a more efficient AAV helps to reduce the required virus concentration. Moreover, it allows the treatment of older (and heavier) specimen, thereby enabling therapies in adults which is usually the stage when subjects develop first symptoms or genetic tests are initiated[45].

Off targets represent a major concern for in vivo gene therapy in patients. To uncover potential off-target editing associated with CRISPR-treatment, WGS is the method of choice because it offers an unbiased analysis of SNVs across the genome in vivo[39]. We showed that after treatment, A > G/T > C nucleotide conversions were not enriched in the hearts that received a high dose of AAVMYO-ABE. This argues against rogue base editing of DNA elsewhere in the genome as it was observed in earlier base editor versions, especially for cytosine base editors[46]. In addition, we did not identify sequences similar to the gRNA near heart-specific SNVs, which indicates that detected SNVs are not related to guide directed activity. Notably, we observed only one missense mutation out of 768 heart-specific SNVs. However, since WGS suffers from sensitivity, it remains unknown whether base editors installed random mutations of low frequency in a subset of cells. Uncovering such edits would require clonal amplification of the target cell type prior to WGS. While such a strategy has been performed for hepatocytes[47], it would not work for cardiomyocytes due to their inability to proliferate. At RNA level, we showed a significant increase from 17 to 19% of A > G edits in one mouse strain treated with 8e-NRCH, which could indicate rogue off-target editing of the base editor. Since we observed this effect only for R636Q HOM mice treated with 8e-NRCH without overt gRNA-dependent effects, the risk for introducing

permanent changes in gene expression is low. Besides off-target edits, base editors are prone to bystander edits in the vicinity of the targeting window[1]. The percentage of bystander edits varied dependent on the deaminase. SpRY and NRCH conjugated to Abemax did not lead to bystander edits whereas NRCH conjugated to hyperactive Abe8e deaminase led to substantial bystander edits of >4% even outside of the putative editing window. Dependent on the target mutation it might make sense to utilize the less active Abemax-ABE version to reduce the danger of bystander edits. In summary, due to the low abundance of random DNA and RNA off-target edits and the reduction of bystander edits by choosing the best ABE, one could argue that the benefit of dramatically decreasing the risk of heart failure outweighs the danger of detrimental off-target editing.

We showed that the extent of *Rbm20* mRNA correction matched well with the number of cardiomyocyte nuclei exhibiting re-localization of RBM20 protein (75%) and the percentage of G-TTN reduction (from 83 to 17%) demonstrating molecular restoration of cardiomyocytes. In addition, we also observed a transcriptional shift towards wild-type in other cardiac cell types such as fibroblasts and epithelial cells upon base editing. SnRNA-seq in *RBM20* patients showed major changes in gene expression and abundance of other cell types besides cardiomyocytes[48], therefore it is encouraging that non-cardiomyocytes also benefitted from the treatment. Overall, our data suggest that base editors prevent the permanent deterioration of heart function. Therefore, we speculate that the positive effect on all cell types is due to lack of structural changes of the heart occurring throughout the live span of the animal. Besides constituting a preventative action for genetically predisposed carriers that have not yet developed DCM, the AAVMYO-ABE treatment may be employed as a curative strategy of adult patients with DCM symptoms.

## Methods

### Ethical Statement

This study was mainly performed in the hybrid mouse strain B6C3F1 backcrossed to C57BL/6J. The animals were maintained in individually ventilated plastic cages (Tecniplast) in an air-conditioned (temperature $22 \pm 2\,°C$, humidity $50 \pm 10\%$) and light-controlled room (illuminated from 07:00 to 19:00). Mice were fed 1318 P autoclavable diet (Altromin, Germany) ad libitum. All experiments were performed with male and female mice except the bulk RNA sequencing which was done only with male mice. Most experiments were performed in at least three mice per condition except for the eYFP titration experiment where only one mouse per AAV concentration was used (Supplementary Fig. 3d, e), the snRNA-seq experiments with only two mice per condition (Fig. 4), and when mice died prematurely before the end of the experiment. The number of mice for all experiments is indicated in the figure legend. For the analysis, the animals were mostly 16 weeks of age except if otherwise indicated in the figure. All animal care and procedures performed in this study conformed to the EMBL guidelines for the Use of Animals in Experiments and were reviewed and approved by the Institutional Animal Care and Use Committee (IACUC).

### Mouse line generation

*Rbm20*-P635L and *Rbm20*-R636Q knock-in mice were generated by zygotic microinjection of recombinant Cas9 (IDT), in vitro reconstituted crRNA:trcrRNA (IDT) targeted to *Rbm20*, and single-stranded donor DNA as a template. The hybrid mouse strain B6C3F1 was used and backcrossed to C57BL/6J for experiments. Sequences of the crRNA and donor template are listed in Supplementary Data 6.

### Cell culture and differentiation

Parental iPSCs and iPSCs harboring the homozygous P633L or R634Q mutation in *RBM20* were previously generated and characterized[24]. Cells were maintained on vitronectin (A31804, ThermoFisher) coated plates with Essential 8™ Flex (A2858501, ThermoFisher) medium and passaged with Versene (15040066, ThermoFisher). Cardiomyocyte differentiation was initiated by addition of 8 µM CHIR99021 (72054, STEMCELL Technologies) in RPMI-1640 medium supplemented with B27 without Insulin (RPMI-Insulin, A1895601, ThermoFisher). After 24 h, 1 volume of RPMI-Insulin was added and after 72 h, medium was changed to RPMI-Insulin with 2 µM Wnt-C59 (5148, Tocris). At day 5 and 7, medium was changed to RPMI-Insulin and at day 9 to RPMI with full B27 supplement (RPMI+Insulin, 17504044, ThermoFisher). At day 11, medium was changed to RPMI+Insulin without glucose and addition of 5 mM DL-lactate. At day 14, RPMI+Insulin was added and at day 16, cells were passaged with TrypLE10x (A1217701, ThermoFisher) and RPMI+Insulin supplemented with 10% knock-out serum replacement (10828028, ThermoFisher) and 1.66 µM Thiazovivin (72252, StemCell Technologies). One day after passaging, the medium was changed to RPMI+Insulin with subsequent medium exchange every 3 days. Passaging was done every 2–3 weeks.

### ABE plasmid cloning

For the transient transfection of base editors in human iPSCs and iPSC-CMs, the following plasmids were used: ABEmax-NRTH (Addgene ID: 136922), ABEmax-NRCH (Addgene ID: 136923), ABEmax-SpRY (Addgene ID: 140003), ABEmax-CP-1041 (Addgene ID: 119808) and ABE8e-CP-1041 (Addgene ID: 138493). Forward and reverse complementary gRNA sequences with compatible overhangs were annealed and ligated with a gRNA expression plasmid (Addgene ID: 53188), which was digested with *BbsI* (R0539S, NEB) prior to ligation. For stable base editor expression, the coding region of Cas9 from the lentiCRISPRv2 plasmid (Addgene: 52961) was replaced with SpRY following a similar strategy as described before[49]. The resulting plasmid was digested with *BsmBI* (R0580S, NEB) and ligated with the annealed gRNA. For the generation of split-intein ABE plasmids, we used Cbh_v5 AAV-ABE N-terminal (Addgene: 137177) and Cbh_v5 AAV-ABE C-terminal (Addgene: 137178) and replaced the coding sequence of SpCas9 with the N- or C-terminal parts of Cas9-NRTH, Cas9-NRCH or Cas9-SpRY using the plasmids from above as template. Moreover, we created a Abe8e-NRCH version by replacing ABEmax on the N-terminal part (common for Cas9-NRTH and Cas9-NRCH) with Abe8e using Addgene plasmid 138489 as template. The C-terminal AAV plasmids were digested with *BsmBI* and ligated with the annealed oligonucleotides encoding the gRNA. Gibson assembly (E2611L, NEB) was used for all cloning assembly steps except for gRNA oligos, which were ligated with the backbone using T4-DNA ligase (M0202L, NEB). Sanger sequencing was performed to validate plasmid assembly and *SmaI* (R0141S, NEB) digestion to monitor the integrity of the ITRs. Guide RNA sequences are listed in Supplementary Data 6.

### Lentivirus production

The lentivirus (generation 3) was produced in Lenti-X 293T cells (632180, Takara) by transfecting the four plasmids with linear PEI (polyethylenimine, 25kD). Virus was collected after 72 h (stored at 4 °C), fresh medium was added and virus was collected again 48 h later. All harvested virus was filtered with 0.45 µm low protein binding/fast flow filter unit. Virus was precipitated using Lenti-X-concentrator (631232, Takara) following the manufacturer's recommendations. Virus was further concentrated by ultracentrifugation with a 20% sucrose cushion at 50,000×*g* for 2 h at 4 °C, and was resuspended in sterile 1× HBSS. Titers were estimated with Lenti-X GoStix Plus (631280, Takara).

### iPSC and iPSC-CM base editing

IPSCs and iPSC-CMs were dissociated one day prior to plasmid transfection as single cells with StemPro™ Accutase™ cell dissociation reagent (A1110501, ThermoFisher) and re-seeded together with RevitaCell™ Supplement (A2644501, ThermoFisher) in 24-well plates

coated with vitronectin. After 24 h, cells were transfected with 375 ng base editor plasmid, 125 ng U6 gRNA plasmid and 100 ng pmax-GFP (Lonza). Lipofectamine™ 3000 or Lipofectamine™ Stem transfection reagent (L3000008 or STEM00008, ThermoFisher) were used for transfection according to the manufacturer's instructions. Medium was changed 1 and 3 days after transfection, and GFP-positive cells were sorted by flow cytometry and analyzed by amplicon sequencing after 25–35 days. Generation of stable SpRY-expressing R636Q iPSCs was achieved by transducing the cells with Lentivirus expressing SpRY and R636Q gRNA2. Cells underwent Puromycin (A1113802, Thermo-Fisher) selection with 2 µg/ml for 14 days before expansion and differentiation to cardiomyocytes. Amplicon sequencing to measure RBM20 editing efficacy was performed before start of the cardiomyocyte differentiation in three independent replicates.

### Recombinant AAV production, purification, and quantification
Recombinant AAVMYO was produced as previously described[50]. Briefly, HEK-293T cells (Stratagene/Agilent) plated on 150 mm dishes were transfected using the 3-plasmid system (pAdH−adenoviral helper function, pRep2cap9myo[22] encoding *rep* and *cap* genes, and transgene plasmid) and PEI. Cells were harvested 3 days later, and viruses were extracted from the cells by four rounds of freeze-thawing. The cell lysates were treated for 1 h with Benzonase to remove non-encapsidated DNA. To remove cell debris, the samples were centrifuged at 4000×*g* and the supernatant was collected. The supernatant was loaded over four layers of iodixanol gradient solution (15, 25, 40 and 60%), followed by a centrifugation for 2.5 h at 183,400×*g* (in average) in a 70Ti rotor. Fractions were collected and those corresponding to the interface of 40 and 60% were pooled, buffer exchanged and concentrated. The viral genome concentration (including in mouse tissue) was determined by ddPCR in a QX200 Droplet Digital PCR System (BioRad), using Taqman primers/probe against the CMV enhancer (Supplementary Data 6), and the purity by silver staining of SDS-PAGE gels.

Recombinant AAV9 was produced in HEK-293T/17 cells (ATCC; CRL-11268) using the triple-transfection method (with linear PEI 25 kDa) in a Corning CellSTACK 5 (CS5). After 72 h, supernatant (600 ml) was collected and stored at 4 °C and 600 ml of fresh medium was added. After an additional 48 h, the first collection was added back to the CS5, and cells were lysed and DNA was degraded by adding Triton X-100 (final concentration of 1%) and 94 µl Benzonase (25–35 U/µl) for 1 h at 37 °C with 100 rpm shaking. The cell debris/virus mix was removed and the CS5 was washed with 200 ml PBS. The washing solution and the cell suspension was centrifuged at 4000 × *g* for 20 min. The supernatant was filtered with a 0.45 µm PES filter and then concentrated to 30 ml using tangential flow filtration. The concentrated virus was then purified by an iodixanol gradient and the titer was determined by qPCR using primers within the CMV promoter.

### Mouse AAV injections
Mice were injected with a mix of AAVs expressing the N-terminal and C-terminal base editor or a *YFP* reporter. Unless otherwise specified, 5e11 vg per AAV were injected in the tail vein of 4-week-old mice. Mice weighted on average 12 g, therefore the total virus concentration injected was 8.33e13 vg/kg. Mice were sacrificed after 6 or 12 weeks and organs collected for subsequent analysis.

### DNA isolation and amplicon sequencing
DNA from human cells was isolated using the Monarch® Genomic DNA Purification Kit (T3010L, NEB) following the manufacturer's instructions and including the recommended RNaseA digestion step. For DNA isolation from mice, the tissue was immersed in 600 µl PBS in tubes containing metallic beads and then processed with a Fastprep homogenizer with two 30 s runs with maximum velocity. One-third of the homogenized tissue was used for DNA isolation with the Monarch® Genomic DNA Purification Kit. No additional Tissue Lysis buffer was added and samples were incubated with 10 µl of Proteinase K for 1 h.

Purified DNA was amplified with human- or mouse-specific primers covering the RBM20 RS-domain mutation hotspot with Nextera-compatible adapters (Supplementary Data 6) using Q5® Hot Start High-Fidelity 2× Master Mix (M0494L, NEB). One microliter of a 1:100 dilution was used for a second PCR attaching sample-specific index barcodes (Nextera XT Index Kit v2 Set A, FC-131-2001, Illumina). Libraries were pooled and cleaned up with 1× AMPure XP beads (A63881, Beckman Coulter) before sequencing with a MiSeq instrument using a 150 bp paired-end run (Illumina).

Demultiplexed amplicons were analyzed using Crispresso2[51] to obtain the frequency of on-target editing and indels and bystander edits with the extended gRNA binding sequence.

### RNA isolation, RT-PCR, qPCR
RNA from human cells was isolated using the Monarch® Total RNA Miniprep Kit (T2010S, NEB) following the manufacturer's instructions and including the on-column DnaseI digestion. For RNA isolation from mice, 1 ml of TRIzol™ (15596026, ThermoFisher) was added to 200 µl of homogenized tissue (see DNA extraction) and processed using the Direct-zol RNA Miniprep Kit (R2052, Zymo Research) with on-column DnaseI digestion. For the heart, RNA was isolated from the left ventricle. 200–500 ng RNA was used as input for the reverse transcription with SuperScript™ IV (18090010, ThermoFisher). For amplicon-seq, RNA was additionally treated with ezDNase™ (11766051, Thermo-Fisher) prior to reverse transcription. RT-PCR or qPCR was performed with 1 µl of 1:2 diluted cDNA using the Q5® Hot Start High-Fidelity 2X Master Mix (M0494L, NEB) or the SYBR™ Green PCR Master Mix (4309155, ThermoFisher), respectively, with gene-specific primers listed in Supplementary Data 6. Delta Ct method using *Gapdh* was used for sample normalization after qPCR. To calculate the fold change, an additional normalization relative to the averaged wild-type RNA expression was performed. RNA copy numbers were determined by ddPCR (see AAV virus quantification) using Taqman primers for the WPRE element (Supplementary Data 6) and Rpp30 (Biorad, assay ID: dMmuCPE5097025) as housekeeping gene.

### Bulk RNA sequencing and analysis
Overall, 500 ng of RNA isolated from left ventricles was processed using the NEBNext® Ultra II Directional RNA Library Prep Kit for Illumina® (E7760L, NEB) with prior enrichment of mRNA by using Oligo dT beads from the NEBNext Poly(A) mRNA Magnetic Isolation Module (E7490L, NEB). After library preparation, the samples were multiplexed (five samples per lane) and sequenced with Illumina NextSeq 2000. For bulk RNA-seq in Fig. 3, samples were processed according to the Smart-seq2 library preparation protocol[52].

Subsequent analysis was performed using a pipeline assembled with Snakemake[53] available at: https://github.com/FerreiraAM/dcm_lgreads_mouse_bulkRNA. The alignment of the different samples was performed using STAR[54]. The GENCODE mouse annotation version vM29 with the primary assembly GRCm39 genome was used. We created the indexes and then aligned the reads for each sample using the default options of the STAR aligner. Number and percentage of mapped reads are shown in Supplementary Data 4. Differential expression analysis was performed with DESeq2[55]. *P* values in Fig. 1d, e, Supplementary Fig. 1e, f, Supplementary Fig. 2, Supplementary Data 1 containing differential gene expression analysis were derived from a one-sided Wald test with adjustments for multiple comparisons. We performed comparisons for each mutation associated with each experiment in R[56] using the count matrices that were created from the BAM files with the Rsubread R package[57]. Log2 fold change (log2FC) per individual was computed for each mutation, using the average per gene from all WT samples of one experiment's mutation: log2FC for

gene A = log2(value of gene A/WT average for gene A). Metascape[58] was used for Gene Ontology analysis.

We used rMATS[59] to detect differential alternative splicing events. Pair-wise comparisons were performed and results from the Junctions Counts (JC) files were analyzed in R. We identified splice junction events that were overlapping between different conditions and filtered for significant events. P values in Fig. 1f, g, Supplementary Fig. 1d, e, and Supplementary Data 2 containing differential splice analysis were derived from a likely-hood ratio test. An event was considered significant if the False Discovery Rate (FDR) was below 0.01 and the average delta PSI-value (relative to WT) was higher than 0.1 or smaller than −0.1. In Fig. 3g, significant splice events were classified in three categories: rescued, mis-spliced or unchanged. We computed the average ΔPSI difference (relative to WT) between PBS- and ABE-treated mice and considered the absolute difference (ΔΔPSI). Events were classified as unchanged if the ΔΔPSI was smaller than 0.1. The remaining events were either classified as rescued or mis-spliced. In addition, we defined the PSI values of PBS-treated samples as the original value ΔPSI_original, the PSI values of base-edited samples as the edited value ΔPSI_edited and used the following criteria:

Rescued:
a. ΔPSI_original > 0 and ΔPSI_edited >= 0 or −0.2 <= ΔPSI_edited <= 0.2 and ΔPSI_original > ΔPSI_edited.
b. ΔPSI_original <0 and ΔPSI_edited <= 0 or −0.2 <= ΔPSI_edited <= 0.2 and ΔPSI_original <ΔPSI_edited.

Mis-spliced:
a. ΔPSI_original > 0 and ΔPSI_edited >= 0 and ΔPSI_original < x_edited.
b. ΔPSI_original <0 and ΔPSI_edited <= 0 and ΔPSI_original > x_edited.
c. ΔPSI_original <0 and ΔPSI_edited >= 0.2.
d. ΔPSI_original > 0 and ΔPSI_edited <= −0.2.

## Mouse cardiomyocyte isolation

For immunostainings of RBM20 granules (Fig. 1a–c), we performed a Langendorff-free isolation of cardiomyocytes as described before[60]. Briefly, the mouse was sacrificed, and the right ventricle was immediately flushed with 7 ml of EDTA-containing buffer. After clamping the ascending aorta, the heart was transferred to a petri dish containing EDTA buffer. Another 10 ml of EDTA buffer was injected in the left ventricle. After injecting 3 ml Perfusion buffer, the heart was transferred to a Petri dish containing collagenase. Subsequently, the left ventricle was injected with 50 ml Collagenase buffer, transferred to a plate with 3 ml Collagenase buffer and cut in small pieces. Overall, 5 ml of stop solution was added and cells were filter through a 100 μm cell strainer and then settled by gravity to enrich for cardiomyocytes. Two rounds of gravity settling were performed before plating the cells on laminin (5 μg/ml in PBS, 23017015, ThermoFisher) coated plates with addition of DMEM/F12 with GlutaMAX™ (10565018, Thermo-Fisher) and 10% FBS. After 2 h, cells were fixed with 4% paraformaldehyde (PFA, methanol-free, 28906, ThermoFisher) for 10 min at RT and stored in PBS for subsequent imaging.

## RBM20 immunostaining and granule quantification

Immunostainings were performed either in isolated adult mouse cardiomyocytes or in tissue sections. Isolated and fixed cardiomyocytes were incubated with 0.5% Triton X-100 in PBS for 5 min before washing with PBS and adding blocking solution (2% BSA diluted in PBS) for 1 h at RT. Cells were incubated overnight in blocking solution containing anti-Rbm20 (PA5-58068, Invitrogen) and anti-sarcomeric alpha-actinin (ab9465, Abcam) both diluted 1:250. Subsequently, cells were washed three times with blocking solution before staining with secondary antibodies Alexa Fluor 488 goat anti-mouse IgG (A11001, Invitrogen) and Alexa Fluor 568 goat anti-rabbit IgG

(A110011, Invitrogen) both diluted 1:1000. Incubation was performed for 1 h at RT before washing three times with blocking solution and mounting the slides in ProLong Gold antifade reagent with DAPI (D1306, Invitrogen). Images were obtained by the LSM 980 AIRY confocal microscope (Zeiss). RBM20 granule quantification in isolated cardiomyocytes was performed by using the ImageJ plugin AggreCount (v1 13)[61], according to the published instructions. The number and the average size of granules per whole cell were used for the figures. At least 10 cells were analyzed for each of three mice per genotype.

For tissue staining, transverse 8 μm sections of heart samples were deparaffinized with xylene and rehydrated to water through decreasing ethanol concentrations. Slide sections were subjected to heat-induced antigen retrieval for 20 min in 10 mM Tris-EDTA pH 9 buffer. Thereafter, these sections were permeabilized with 0.3% Triton X-100, blocked in 5% donkey serum and incubated overnight at 4 °C with a rabbit anti-Rbm20 antibody (PA5-53068, Invitrogen) at 0.5 μg/ml. Immunofluorescent detection was done with a tyramide signal amplification using an anti-rabbit-HRP (12–348, Sigma), Biotinyl-tyramide (SML2135, Sigma) and streptavidin-Alexa 488 (S11223, Molecular Probes). Images were acquired by widefield microscopy with an automated whole slide scanner. Nuclear versus cytoplasmic RBM20 localization was quantified manually in 3–4 slices per mouse heart for two mice per condition and a total of 250–500 cells.

## PicoSirius Red staining

Hearts were processed for standard paraffin embedding. Sagittal sections were collected around the mid portion of each sample at 8 μm onto Superfrost Plus slides. Following deparaffinization and hydration with alcohols to water, the sections were stained with a solution of picrosirius red (0.5 g/500 ml saturated picric acid; Sigma) for 1 h at RT. Sections were then washed in two changes of acidified water (5 ml Glacial acetic acid/1 later distilled water), dehydrated in 100% ethanol and mounted in Permount. Images were acquired with an automated whole slide scanner.

## Echocardiography

Mice were anesthetized using 2–2.5% isoflurane (HDG9623V, Baxter Deutschland GmbH, Germany), while heart and respiratory rate were continually monitored. Cardiac echocardiography was performed using a Vevo 2100 Imaging System with a MS400 Transducer (both FUJIFILM VisualSonics, Inc., Canada), and B-mode and M-mode images of short axis and long axis were taken. During echocardiography, mice were placed on a heating pad to avoid a decrease in body temperature. Echocardiographic parameters were then analyzed using the Visual-Sonics VevoLab software.

## Vertical SDS agarose gel electrophoresis (VAGE)

VAGE for detection of TTN protein isoforms was performed as described before[62] with minor modifications. A piece of 5–10 mg heart tissue was lysed in 40 volumes (w/v) of VAGE sample buffer (8 M urea, 2 M thiourea, 3% SDS, 0.03% bromphenol blue, 0.05 M Tris-HCl, 75 mM DTT, pH 6.8) with a pistel in a microtube at 60 °C for 2–3 min. Subsequently, 50% glycerol buffer (50 ml H$_2$O, 50 ml Ultrapure Glycerol, 1 Tablet protease inhibitor cocktail (11697498001, Roche)) was added (final concentration 12%) and the samples were processed for another 3–5 min at RT. After a cooling period of 5 min on ice, a centrifugation for 5 min at 16,000× g occurred. The supernatant was taken and stored at 80 °C. The samples were thawed by heating to 60 °C for 2 min and analyzed using VAGE. After the gel run, the gel was fixed for 1 h in 50% methanol, 12% acetic acid 5% glycerol in ddH$_2$O and dried overnight. The gel was rehydrated in H$_2$O, stained with Coomassie and scanned for quantification. Analysis and quantification was performed with the AIDA software.

## Whole-genome sequencing and analysis

DNA for WGS was prepared by PCR-free library preparation according to the NEBNext Ultra II DNA PCR-free Library Prep kit (E7410L, NEB). Sequencing was performed by Illumina NextSeq 2000 P3 150PE.

Analysis of the sequencing data was performed using a customized Snakemake workflow. Raw FASTQ files were initially processed for 3' adapter trimming from raw FASTQ files using cutadapt v.3.5[63]. The trimmed reads were subsequently aligned by bwa-mem v.0.7.17[64] to a hybrid reference sequence of mouse genome mm10 concatenated with AAV vector backbone sequences comprising the N- and C-terminal components of SpRY-Cas9-ABE. Number and percentage of mapped reads are shown in Supplementary Data 4. The BAM files were sorted, marked for PCR duplicates and recalibrated for base quality scores using GATK v4.1.9.0. Three variant callers were applied for SNP (GATK Mutect2 v.4.1.9.0 (MU)[65] GATK HaplotypeCaller v.4.1.9.0 (HC)[65], Lofreq v.2.1.5 (LF)[66]) and Indel (MU, HC, Scalpel v.0.5.4 (SC)[67]) calling, respectively. For MU and HC, variant calling was performed in cohort mode using BAM files of all three tissue samples from the same individual. Therefore, allelic depth and frequency (AF) of reference and alternative alleles were recorded even for variants not present across all tissue types. For LF and SC, the default parameters were used to call variants from a single sample at a time. All variants were left-aligned and normalized using bcftools (v.1.9)[68] to allow a comparison between variant callers. For further analysis, the allelic depth called by MU was used if present, and was otherwise replaced by values determined by HC. ANNOVAR (v.2020-06-08)[69] was used to add functional annotations to the detected variants. To identify variants with high confidence, we required a variant to (1) be called by at least two variant callers, (2) be covered by at least five reads per tissue type, and (3) have at least two alternative allele reads across all tissues. After this quality filtering step, tissue-overlapping variants were defined as variants present in all three tissues. To identify novel mutations with high confidence, variants overlapping with any known variant annotated by the mouse genome project[70] or dbSNP[71] were excluded. From this pool of variants, tissue-specific variants were defined as variants with an AF > 0 in the tissue in question and an AF = 0 or non-measured variants in the other tissues. To further characterize tissue-specific variants, they were examined for a potential causation by the CRISPR base-editor treatment. For each variant, a section of ±30 bases around its start site was investigated for sequence homology to the gRNA and PAM sequence. A custom script was developed for the sequence alignment and calculation of minimum edit distance, while only allowing 1 bp indels or mismatches in the seed region but not in the PAM site.

## RNA-seq variant analysis

Analysis of the sequencing data was performed using a customized Snakemake workflow. Raw FASTQ files were aligned by STAR v2.7.9a in 2-pass mode[54] to the same hybrid reference sequence used in the WGS analysis. The BAM files were sorted and marked for PCR duplicates using GATK v4.1.9.064. Three variant callers were applied for variant calling: (GATK HaplotypeCaller v.4.1.9.0 (HC)[65], Strelka v.2.9.10 (ST)[72] and Platypus v.0.8.1 (PL))[73]. For HC, the sorted and marked reads were pre-processed as described in the GATK Best Practices for RNA-seq variant calling[74,75]. For PL, sorted and marked reads were processed with Opossum v.0.2[76] and ST used sorted and marked reads as their input and was run in RNA mode. All algorithms called variants in cohort mode using BAM files of all three tissue samples from the same individual. All variants were normalized, left-aligned, and annotated as described in the WGS analysis section. For further analysis, the allelic depth called by PL was used if present, and was otherwise replaced by values determined by HC. To identify variants with high confidence, we required a variant to (1) be called by at least two out of three variant callers, (2) be covered by at least five reads (tissue were investigated individually), (3) have at least two alternative allele reads across all tissues, and 4) be located in exons, introns, or UTR3/5 regions. Known variants annotated by dbSNP or MGP were excluded, and the remaining variants were grouped into tissue-specific or tissue-overlapping variants.

For Supplementary Fig. 7b, REDItools2[77] was utilized to extract all reads from the target region. Triplicate reads were summed up, and the fraction per base and position was calculated. Investigation of sequence similarity between gRNA regions surrounding the SNVs was performed as described in the WGS analysis.

## Nuclei isolation and snRNA-seq

Nuclei from mouse hearts were isolated using previous protocols with some adaptations. Briefly, hearts were washed three times with PBS, minced and incubated with 5 ml 1× Red Blood Cell Lysis Buffer (Z3141, Promega) for 5 min with manual shaking. In total, 5 ml PBS was added, followed by centrifugation at $500 \times g/2$ min and an additional washing step with 10 ml PBS. The pellet was resuspended in 1 ml homogenization buffer[78], and dounced 8× with pestle 'A' and 20× with pestle 'B' on ice. Nuclei were filtered through a 70 μM strainer, followed by a 40 μM strainer and a 20 μM strainer. Nuclei were centrifuged at $1000 \times g/5$ min and pellet resuspended in 2 ml homogenization buffer. Nuclei solution was layered on 10 ml sucrose buffer[79] and centrifuged $1000 \times g/5$ min. The pellet was washed with 2 ml homogenization buffer and resuspended in 0.2 ml PBS (calcium and magnesiumfree) with 2% BSA and 0.2 U/μl RNasin® Plus RNase inhibitor (N2615, Promega). Nuclei stained with Dapi were sorted by flow cytometry in FACS buffer. The gating strategy is depicted in the source data. Sequencing libraries were prepared with the Single Cell 5' Reagent Kit v2 Dual Index (1000265, 10xGenomics) and sequencing was performed by NextSeq550 Mid 75 PE.

## SnRNA-seq analysis

SnRNA-seq data was aligned to the mouse reference mm10 (GENCODE vM23/Ensembl 98) using 10x Genomics` Cell Ranger 7.0. Downstream analysis on the gene count matrix was performed in R v4.2.1 and Seurat v4. At the pre-processing stage, the cells were filtered such that each cell has between 100 and 2500 active genes with non-zero counts. Cells exhibiting more than 1% counts belonging to mitochondrial genes were not included. The counts of each cell were log-normalized, and the 2000 most variable features were identified in each run separately. Pre-processed data from different runs were harmonized using the FindIntegrationAnchors method in Seurat. Principal Component Analysis (PCA) was performed on the integrated data to identify the 30 largest contributors to gene expression profile variation. Clusters were identified using the Louvain algorithm[80] with resolution parameter 0.5. For visualization of the cell clusters, Uniform Manifold Approximation and Projection (UMAP) reduction[81] was performed on the 30 PCs. Each cluster was mapped to a specific cell type using markers provided by the Heart Cell Atlas[37]. Within each cluster, an additional PCA was performed on the pre-integration gene expression data to identify sub-cluster variation. UMAP visualization was obtained from the 5 largest PCs, while pairwise cell distances were calculated from the first two PCs using the Euclidean metric.

For cell type-specific comparison of inter-genotype variations, differential expression analysis was performed between the WT and P635L HOM cells within each cluster using the Wilcoxon Rank Sum test, with maximum $P$ value 0.05. Per cell type, up to 15 up- and downregulated genes were identified (Supplementary Data 3). Their expression values were rescaled across cells such that their values were between 0 and 1, where 1 indicates maximum activation of the gene in a cell, and 0 indicates non-expression. An overall activation score was then calculated for each cell by averaging its rescaled expression values across the list of up-/downregulated genes. By scanning a threshold value between 0 and 1, the percentage of cells with activity scores above threshold was then used as a proxy for that population's expression of a set of genes. The critical threshold for comparison was chosen as the value where the genotype with downregulated activity drops below 50% active cells.

## Statistics & reproducibility

GraphPad Prism software (v9.3.1) was used for statistical analysis except RNA-seq data which was analyzed in R. Data were analyzed with unpaired *t* tests, one-way or two-way ANOVA with Tukey's multiple comparison posttest or log-rank test. Name of the test, *P* value and number of biological replicates are indicated in each figure legend. Data are displayed as means ± SEM. No statistical method was used to predetermine sample size. No data were excluded from the analyses. The experiments were not randomized, and the investigators were in most cases not blinded to allocation during experiments and outcome assessment.

## Reporting summary

Further information on research design is available in the Nature Portfolio Reporting Summary linked to this article.

## Data availability

The main data is available in the main text or the supplementary materials. WGS data was uploaded to the SRA database at NCBI under the accession code SRP415422: Bulk RNA-seq data was uploaded to NCBI GEO under the accession code GSE226130: Single nuclei RNA-seq data was uploaded to NCBI GEO under the accession code PRJNA960908: WGS and SnRNA-seq data was aligned to the mouse reference mm10 (GENCODE vM23/Ensembl 98), for bulk RNA-seq analysis, the GENCODE mouse annotation version vM29 with the primary assembly GRCm39 genome was used. Supplementary Data 1, 2, 3 and 5 containing the list of differentially expressed (bulk and single-nuclei) and spliced genes, as well as the list of tissue-specific variants has been made public on Figshare: https://figshare.com/projects/Striated_muscle-specific_base_editing_enables_correction_of_mutations_causing_dilated_cardiomyopathy/156347. Source data are provided with this paper.

## Code availability

Data were analyzed with GraphPad Prism software (v9.3.1) and R (v4.2.2). DEseq2 (v3.17) and rMATS (v4.1.2) were used for the analysis of bulk RNA-seq. RNA variants were called with GATK HaplotypeCaller v.4.1.9.0, Strelka v.2.9.10 and Platypus v.0.8.1 and reads were processed with Opossum v.0.2. Metascape (v3.5.20230501) was used for Gene Ontology analysis. For the WGS analysis, we used cutadapt (v.3.5), bwa-mem (v.0.7.17), GATK Mutect2 v4.1.9.0, GATK HaplotypeCaller v.4.1.9.0, Lofreq v.2.1.5, Scalpel v.0.5.4 and bcftools (v.1.9. SnRNA-seq data was aligned to the mouse reference mm10 (GENCODE vM23/Ensembl 98) using 10x Cell Ranger 7.0. Downstream analysis on the gene count matrix was performed in R v4.2.1 and Seurat v4. Details are provided in 'Methods' section. A description of the analysis pipeline for RNA-seq and WGS can be found on GitHub: https://github.com/FerreiraAM/dcm_lgreads_mouse_bulkRNA. https://github.com/LeonieKuechenhoff/wgs_variant_analysis_RBM20.

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

## Acknowledgements

We sincerely acknowledge the Laboratory Animal Resources (LAR) facility at EMBL for supporting the mouse experiments. Specifically, we thank Ernesto de la Cueva Bueno who supported ethics approval for the mouse experiments, Frank Diego Montoya Castillo who performed AAV injections and organ harvesting, and Isabel Clara Rollan Delgado who performed the zygotic injections for generation of the *Rbm20* knock-in mice. Guide RNAs and donor sequences for the mouse line generation were designed by Anya Grozhik. The students Anastasiia Korosteleva, Maral Azodi and Stephanie Wendel performed supporting experiments related to this study. Further support was received from the Genomics Core Facility at EMBL, headed by Vladimir Benes, for RNA and DNA sequencing. We are very grateful for the expertise of Laura Villacorta, Mireia Osuna Lopez and Hilal Ozgur in generating WGS and bulk RNA sequencing libraries, as well as performing the Illumina sequencing. AAV9 and lentivirus were prepared by the Genetic and Viral Engineering Facility at EMBL Rome headed by Jim Sawitzke. Moreover, we acknowledge Carmen Judis (MDC Berlin) for supporting the TTN VAGE analysis. This work was mainly funded by the Bundesministerium für Bildung und Forschung (BMBF), the Else Kröner-Fresenius-Stiftung (EKFS), the CRC 1550 "Molecular Circuits of Heart Disease" and the Leducq Foundation. M.G. has received funding from the European Union's Horizon 2020 research and innovation program under the Marie Sklodowska-Curie grant agreement No. 101031265.

## Author contributions

M.G. and L.M.S. conceived the project. M.G., L.S., J.K., S.C., and E.P. performed experiments. A.C., L.K., A.-M.F., S.L. and C.D. analyzed the data. K.R. and C.K. were responsible for AAVMYO generation with guidance from D.G. M.H.R. analyzed TTN protein splicing with input from M.Go. M.H. performed mouse echocardiography with support from J.B. M.G. and L.M.S. wrote the manuscript. All authors commented on the manuscript.

## Funding

## Competing interests

The authors Markus Grosch and Lars Steinmetz filed an invention disclose describing the combination of AAV and base editors for treatment of hereditary dilated cardiomyopathy (U.S. Provisional Patent Application No. 63/423,716, status: filed: November 8, 2022). The remaining authors declare no competing interests.

## Additional information

[1]European Molecular Biology Laboratory (EMBL), Genome Biology Unit, Heidelberg, Germany. [2]Department of Genetics, Stanford University School of Medicine, Stanford, CA, USA. [3]DZHK (German Center for Cardiovascular Research), Partner Site Heidelberg/Mannheim, Heidelberg, Germany. [4]Klaus Tschira Institute for Integrative Computational Cardiology, University of Heidelberg, Heidelberg, Germany. [5]Department of Infectious Diseases/Virology, Section Viral Vector Technologies, Medical Faculty, BioQuant, University of Heidelberg, Heidelberg, Germany. [6]Translational Cardiology and Functional Genomics, Max Delbrück Center for Molecular Medicine in the Helmholtz Association, Berlin, Germany. [7]German Center for Cardiovascular Research (DZHK), Partner Site Berlin, Berlin, Germany. [8]Epigenetics and Neurobiology Unit, EMBL Rome, Monterotondo, Italy. [9]Institute of Experimental Cardiology, University Hospital Heidelberg, Heidelberg, Germany. [10]Department of Cardiology, Charité Universitätsmedizin Berlin, Berlin, Germany. [11]German Center for Infection Research (DZIF), Partner Site Heidelberg, Heidelberg, Germany. [12]Stanford Genome Technology Center, Palo Alto, CA, USA. ✉e-mail: Lars.Steinmetz@stanford.edu

