## [Peer Review File · Nature Communications]

Reviewers' Comments:

Reviewer #1:

Remarks to the Author:

Dilated cardiomyopathy (DCM) is the second most common cause for heart failure, and approximately 30% of DCM patients harbor heritable mutations which are amenable to CRISPR-based gene therapy. In a previous study, Dirk Grimm lab discovered a peptide-displaying AAV9 mutant called AAVMYO that exhibits superior efficiency and specificity in the musculature including skeletal muscle, heart and diaphragm. In this manuscript, Markus Grosch et al leveraged for the first time AAVMYO for systemic delivery of base editors to cardiomyocytes, and repair two pathogenic mutations in Rbm20's RS-domain resulting in near-complete rescue of the disease phenotype in mice with no evidence for gRNA-dependent off-target activity. The job was great and meaningful. However, this manuscript suffers from several issues that need to be addressed by the authors.

Major points

- 1, Rbm20 RS-domain mutation were recently shown to result in aberrant formation of cytoplasmic granules, and amplified the disease phenotype, but P635L HOM and R636Q HOM mice were no significant worsening of the DCM-associated phenotype, and survival curves were lesser extent than other Rbm20 RS-domain mutations. The result seems inconsistent with the previous study (PMID: 36417486). Why, a fuller discussion is also needed.
- 2, Fig. 2g showed that 2.7% of bystander edits in vivo. Whether the bystander edits belongs to RS-domain mutations? and whether it effected the heart function.
- 3, To evaluate the extent of phenotype rescue upon base editing, the survival curves needed to be provided in fig.3.
- 4, Authors should increase the number of animals to support the results that LVID and cardiac volume decreased upon base editing (Line 289-291).
- 5, AAV mediated treatment strategies are known for their toxic side-effects. Did the authors look into serum biochemical markers like ALT, total protein, urea etc.?
- 6, Authors need to perform experiments to detect potential host immune response, such as IL2, IL15 or AAV antibody to confirmed the results in Line 327.
- 7, The off-target effect of ABE system is a topic of concern, and WGS was greatly influenced by the genetic background of the individual animal. Sequence analysis of the candidate off-target amplicons was needed to reveal no notable editing at any of the top 8-15 tested off-target sites.

Minor points

- 1, Line73-74 "3% of patients with aggressive, early onset DCM" should provide reference.
- 2, Line98-99 Did mice carried both P635L and R636Q mutations? Sentences needed to be more accurate.
- 3, The position of bystander need to be clearly in Sup. Fig. 3b etc.
- 4, The variant of number of active genes, total transcript counts and percentage of mitochondrial gene counts per cell were large (Sup fig5.a-c). Please provide the sequencing quality results and sample numbers in the supplementary.
- 5, AAVMYO exhibited superior efficiency and specificity in the musculature and liver detargeting. It was amazing and interested. Although previous work (PMID: 33116134) have been reported, author also needed to provide more detailed information about AAVMYO sequence or mutants in supplementary.
- 6, The vertical axis in the figure were confusing and did not told whether editing efficiency was at the DNA or RNA level, such as fig2. g, h and i. It should be shown clearly.
- 7, The figure legend should be clear about the animals numbers and tissues.

Reviewer #2:

Remarks to the Author:

In this study, Grosch, et al generated Rbm20 knock-in mice resembling human RBM20 cardiomyopathy and established base-editing therapy using their muscle-specific AAVMYO vector combined with a variety of base-editing constructs. The authors demonstrated that base-editing therapy repaired genomic variant both in in vitro and in vivo with high efficiency, corrected transcriptional profiles and abnormal cytoplasmic localization of Rbm20 protein, and finally

improved physiological phenotypes. Furthermore, the authors proved that aberrant off-target editing was not observed using whole-genome sequencing. This study presents comprehensive experimental data based on step-by-step robust pilot experiments and provides a proof of concept of base-editing therapy for future clinical application. I have some minor comments.

1. In Sup. Figure 1g, mRNA expression levels of Col1a2 and Mmp2 are more than 2-fold increased from WT, although significant difference was not labeled. Is this correct?

2. The authors stated the increased LVID in knock-in mouse models (line 136), but there is no significant difference in LVID among genotypes in in Sup. Figure 1k.

3. Time dependent increase in the editing efficiency both in genomic DNA and cDNA shown in Sup. Figure 3h and Figure 2i (the efficiency of in vivo base editing was approximately 2-fold increased at 12 weeks compared to 6 weeks after injection) is interesting. Please explain the putative underlying mechanism in discussion.

4. The highest levels of in vivo editing efficiency were achieved by high cell tropism of AAVMYO that only infects cardiomyocytes, as authors stated (line 231). Cell-type specific expression profiling of the viral transcripts derived from the AAVMYO-specific components using the authors' snRNA-seq data may further strengthen the beneficial effects of AAVMYO-based therapy.

Reviewer #3:

Remarks to the Author:

The manuscript by Grosch et al examines the ability of an adenoviral gene transfer vector to target CRISPR agents in a tissue-specific manner to the heart/skeletal musculature. Previous investigations by the authors and others established that mutations in Rbm20, a factor that regulates splice-site choice during alternative splicing, can lead to a dilated cardiomyopathy (DCM) phenotype. Transgenic mice were developed encoding the P635L and R636Q mutations which reflect RBM20 mutations in DCM patients. In addition to the production of aberrantly spliced transcripts, cytoplasmic granules containing RBM20 were also found in the cardiomyocytes of these mice. Using knock-in mice of the P635L and R636Q RBM20 gene, the authors utilized an adeno-associated viral vector which specifically targets striated muscle to deliver and express CRISPR reagents that edited the RBM20 mutations to wild-type sequences. These edited sequences restored cardiac function and the transcriptional profile of wild-type hearts and cardiomyocytes. Delivering tissue-specific reagents for gene editing in the whole animal has been problematic in the past. The methods employed by these investigators address this problem and are logical extensions of their previous work in furthering gene editing processes for the correction and treatment of mutations associated with DCM, specifically the splicing factor gene RBM20. Overall, this is a very well-conceived and scientifically-sound investigation that integrates newly developing gene-editing techniques with potential for treatment of cardiovascular diseases. The research and methodology presented in this manuscript provides a comprehensive example of cardiomyocyte specific gene targeting to correct DCM mutations and analyses of the molecular, physiological, and cytological effects in isolated cardiomyocytes, iPSCs, and in vivo mice. The experiments are well-designed, with proper controls and clearly presented results. The discussion of the work is well integrated into the current state-of-knowledge in this field.

There are, however, several minor suggestions that would strengthen the manuscript.

1. Lines 97 – 99: What is the amino acid identity between human and mouse RBM20 sequences? Also, a discussion of the changes on the protein structure and function that occur with the P635L and R636Q mutations (and orthologous changes in DCM patients) may enhance the understanding of why and how these mutations affect the alternative-splicing process in various genes.

2. Lines 118 – 119: There should be additional discussion of why there was little overlap between P635L and R36Q DEGs and DSGs.

3. Lines 120 – 122: In Fig. 1g, some genes (i.e. Ttn) are listed more than once – why?

4. Line 136 “increased cardiac... (LVID)”: In Supp. Fig 1j, k – there appears to only be a significant change in cardiac volume in the P635L Homo samples, not in the other genotypes (Supp. Fig 1j). As shown, there are no changes in the LVID for any genotypes in Supp. Fig 1k. The text in the manuscript should be changed to reflect this.

5. Lines 186 - 187: the wording of "partially approached" is awkward – this should be change. Also, in Fig 2e, the error bar for the WTd32 sample is very large – can this be tightened up to ascertain whether there was statistical significance among the samples?

6. Line 196: Please provide the rationale for selecting the P635L HOM mice for further testing and analyses. The P636Q mice displayed a more severe phenotype. Why not select both Het and HOM mice of both genotypes?

Summary of revisions

We thank the reviewers for their helpful suggestions and have now addressed their concerns. To briefly summarize our main revisions before our point-by-point responses below, in particular, we have expanded our off-target analysis by performing targeted amplicon-seq and integrating variants detected at the RNA level (**Sup. Fig 6f** and **Sup. Fig. 7**). Besides, we added more *in vivo* data by phenotyping of one-year-old Rbm20 R636Q mutant mice which showed an increase in left ventricle internal diameter (LVID) and cardiac volume, consistent with previous studies (**Sup. Fig 1m, n**). We also performed echocardiography in mice injected with AAVMYO containing an hTNNT2-promoter driven base editor cassette and demonstrated significant improvement in heart pump function (**Sup. Fig 4g**) which shows that AAVMYO-selectivity can be further improved by combination with cardiac promoters. Based on reviewers' suggestion, we also analyzed base editor expression from our single-nuclei RNA-seq data and validated their expression in cardiomyocytes which confirmed the muscle tropism of AAVMYO on a single-cell/nuclei level. In all, these reviewer's suggestions greatly improved the quality of the manuscript. We are grateful for the constructive feedback by all reviewers. We address the reviewers' comments individually below (reviewers' comments in blue italics and our responses in black font; text in green indicates newly added text in the manuscript).

Reviewer #1 (Remarks to the Author):

Dilated cardiomyopathy (DCM) is the second most common cause for heart failure, and approximately 30% of DCM patients harbor heritable mutations which are amenable to CRISPR-based gene therapy. In a previous study, Dirk Grimm lab discovered a peptide-displaying AAV9 mutant called AAVMYO that exhibits superior efficiency and specificity in the musculature including skeletal muscle, heart and diaphragm. In this manuscript, Markus Grosch et al leveraged for the first time AAVMYO for systemic delivery of base editors to cardiomyocytes, and repair two pathogenic mutations in Rbm20's RS-domain resulting in near-complete rescue of the disease phenotype in mice with no evidence for gRNA-dependent off-target activity. The job was great and meaningful. However, this manuscript suffers from several issues that need to be addressed by the authors.

We thank the reviewer for the insightful and detailed comments regarding our manuscript. We have discussed all points below and integrated further analyses into the manuscript that expand the scope of off-target analysis. Moreover, based on these suggestions we have modified the figures to highlight the position of bystander edits, to clarify if RNA and DNA editing was analysed and to provide a table to describe the mapping statistics of the snRNA-seq experiments. Furthermore, we reached out to the group of Eric Olsen and discussed observed differences in survival rate of our mutant mice.

Major points

1, Rbm20 RS-domain mutation were recently shown to result in aberrant formation of cytoplasmic granules, and amplified the disease phenotype, but P635L HOM and R636Q HOM mice were no significant worsening of the DCM-associated phenotype, and survival curves were lesser extent than other Rbm20 RS-domain mutations. The result seems inconsistent with the previous study (PMID: 36417486). Why, a fuller discussion is also needed.

We thank the reviewer for this observation. We speculate that the reason might be the genetic background of the animals. Our mice were backcrossed to C57BL/6J whereas the animals in the Olsen study had a C57BL/6N background. We reached out to these authors to confirm. There

was a study comparing both mouse strains upon cardiac stress induced by transverse aortic constriction (TAC) which found that they exhibit very different response to TAC stimulation with BL/6N being more susceptible to eccentric hypertrophy and age/time-dependent deterioration of cardiac function (PMID: 32699840). Another study comparing different phenotypic aspects of 6N and 6J mice, also found stark differences in heart weight, pulse rate and systolic arterial pressure which could influence cardiovascular function in these strains (PMID: 23902802). In summary, there is evidence showing differences in heart function between 6J and 6N, which could explain why our mice are less likely to spontaneously die from heart failure. We clarified this issue in the result section:

L136-39: "...Notably, we backcrossed our mutant mice to C57BL/6J where others have used C57BL/6N³⁰, which could explain the differences in the survival as C57BL/6N is more susceptible to cardiac deterioration upon pressure overload³¹..."

2, Fig. 2g showed that 2.7% of bystander edits in vivo. Whether the bystander edits belongs to RS-domain mutations? and whether it effected the heart function.

We thank the reviewer for this comment based on which we have extensively modified our analysis of bystander edits. Bystander edits are discussed at two sections in the manuscript, namely for Fig. 2 when different base editors/gRNA combinations were evaluated (section 1), and for Fig. 3, when the base editors 8e-NRCH and SpRY were used for long-term editing of 12 weeks (section 2).

Section 1:

The main bystander edit (T_2) for 8e-NRCH introduces a synonymous change of the CGT codon to a CGC codon both encoding arginine. We clarified this in the text and modified Fig. 2g which shows editing events analyzed after 6 weeks (see below). For editing using gRNA1, we observed a second bystander edit, named T_1 , occurring in 0.20-1.31% of reads, changing the codon from TCT (serine) to CCT (proline) which could potentially impair the function of RBM20. This gRNA was subsequently discontinued and only gRNA2 was used for 12-week editing. In the new Fig. 2g, we disentangled bystander edits by their exact position within the RS-codon sequence. Besides, we have added a cartoon showing the position of the bystander edits in Sup, Fig. 3b, c and Sup. Fig. 4b as response to the reviewer's minor comment 3. Moreover, we specified in the axis label whether we analyze Rbm20 DNA or RNA editing, as response to the reviewer's minor comment 6.

Section 1:

L217-25: "...We also tested NRCH conjugated with the latest and most efficient version of adenine deaminase, namely Abe8e37 (referred to as 8e-NRCH), and observed the highest editing with 21.4% on average (Fig. 2g). This editor, however, also showed bystander edits of 2.7%, the most common bystander edit of which (T_2 ; 2.64%) introduces a synonymous codon change and is likely inconsequential. Of note, due to different positioning of the base editor, a second non-synonymous bystander edit (T_1) was observed for gRNA1 in up to 1.31% of reads leading to a codon change from TCT (serine) to CCT (proline). Therefore, the use of gRNA1 was discontinued for subsequent long-term editing and phenotyping. T_1 was also detected in 8e-NRCH combined with gRNA1 but only in 0.09% of reads on average..."

New Fig. 2g: Percentage of editing of P635L HOM mice injected with AAVMYO carrying different gRNA-base editor combinations or PBS as empty control. For NRCH-gRNA1, AA9 was also used as vector. N = 2-7 mice per condition. Significance was assessed using unpaired t-tests *** p < 0.001, ** p < 0.01, * p < 0.05. Sequence shows location of the on-target edit in blue and the two bystander edits in red. Numbers depict the position of the nucleotides within the targeting gRNA (gRNA2 was used as reference) with the PAM sequence in position 21-23.

Section2:

In Fig. 4 and Sup. Fig. 5, we analyze the editing outcome 12 weeks after treatment with AAVMYO-ABE. For gRNA2 combined with SpRY, we do not observe any bystander edits. For gRNA2 combined with 8e-NRCH, we observed the synonymous bystander edits T₂ (4.09% of reads in average), T₋₂ (0.20%) and T₁₇ (0.42%). Moreover, the missense mutation T₁ was observed in 0.33% of reads in average. We have clarified this in the figure caption in Sup. Fig. 4b and in expanded this section in the results. We also provide another supplementary figure (Sup. Fig. 4c) showing bystander edits in only reads that have also received the correct edit which indicated that only repaired alleles were prone to bystander edits.

Section2:

L288-94: "...Compared to editing after 6 weeks, overall, more bystander edits were detected in P635L mice treated with 8e-NRCH. The main synonymous bystander edit T₂ occurred in 4.09% of reads on average followed by a missense mutation T₁ (0.33%). Two other synonymous mutations T₋₂ (0.20%) and T₁₇ (0.42%) were observed (Sup. Fig. 4b). R636Q mice treated with 8e-NRCH exhibited one bystander edit A₂ (0.69%). Notably, we observed bystander edits only in reads that have also received the correct edit indicating that only repaired alleles were prone to bystander edits, which effectively lowers the editing efficacy (Sup. Fig. 4c)..."

New Sup. Fig. 4b: Allele frequency of bystander edits in mice treated with AAVMYO-ABE. N = 3-4 mice per condition. Sequence shows location of the on-target edit in blue and the bystander edits in red. Numbers depict the position of the nucleotides within the targeting gRNA with the PAM sequence in position 21-23.

New Sup. Fig. 4c: Example allele frequency determined by Crispresso2 for one mouse treated with AAVMYO harboring the base editor 8e-NRCH and gRNA2. On-target location is indicated in blue and location of observed bystander edits in red.

3, To evaluate the extent of phenotype rescue upon base editing, the survival curves needed to be provided in fig.3.

We thank the reviewer for this suggestion. The differences in survival between our mutant mouse strains and wildtype are not big enough (potential reasons are discussed in response 1) to serve as quantitative assessment for the efficacy of base editor-mediated gene repair. Only when assaying >20 mice, statistically significant differences were observed in overall survival (**Fig. 1j**) however, we could not analyze so many mice treated with AAVMYO due to the scarcity of the virus. We indeed do not observe any sudden death after treatment and follow-up for 12 weeks, however, mice injected with the carrier solution PBS did also not die prematurely during the observation period. Therefore, we relied on molecular parameters such as splicing and gene expression as well as assessing more specifically the heart pump function by echocardiography to assess phenotype improvements after base editing.

4, Authors should increase the number of animals to support the results that LVID and cardiac volume decreased upon base editing (Line 289-291).

Similar to the probability of survival (discussed in response 3), the dynamic range of LVID between mutant mice and wildtype was not big enough to allow quantitative statements about the efficacy of base editing (See Sup. Fig. 1k). The cardiac volume was significantly changed in P635L HOM mice relative to WT (See Sup. Fig. 1j), however since this parameter correlates with the overall size of the animal, it is strongly subjected to biological variation. For this reason, we decided to rely on other parameters such as splicing, granule formation and ejection fraction to determine the efficacy of base editing-mediated phenotypic improvement. While we see a trend in improved LVID and cardiac volume, we are pointing out that it is not yet significant:

L305-7: "... In line with the restoration of cardiac function, LVID and cardiac volume decreased upon base editing albeit without reaching statistical significance (**Fig. 3i, j**)..."

5, AAV mediated treatment strategies are known for their toxic side-effects. Did the authors look into serum biochemical markers like ALT, total protein, urea etc.?

We appreciate this reviewer's comment which touches a crucial topics towards clinical application of the base editor treatment. AAVs have been used extensively in small and large animal models, as well as in humans, owing, amongst others, to their low immunogenicity. However, as the reviewer has pointed out, there have been reports of severe adverse effects leading in some cases, unfortunately, to deaths, related to high dose. These adverse effects include hepatotoxicity (increase in liver transaminases), dorsal root ganglia toxicity and thrombotic microangiopathy

amongst others. They can be attributed to the hyperactivation of the innate immune system, activation of the complement system or of cellular immunity (reviewed here: PMIDs: 36715794, 34777364). These severe adverse effects have been observed in large animal models, such as non-human primates or piglets and are associated with dosages of 1×10^{14} vg/kg (PMID: 29378426, 33177182). In smaller animal models like mice, an activation of immune pathways in the liver was observed at high dosage (6×10^{13} vg/kg) of a liver targeted serotype, 5, at 12 weeks post injection, and a loss of genome copies was observed after this time-point⁵. In our study, we used a slightly higher dosage ($\sim 8 \times 10^{13}$ vg/kg) but our serotype is liver-detargeted which likely reduces the chance of liver toxicity. The main problem is that AAV-induced immunogenicity varies drastically between species, which is e.g. discussed in the following review stating: “The many years of encouraging safety data from numerous animals and humans treated with rAAV suggests that the malignancy risk for humans might not be best predicted by experience with rodents” (PMID: 26906613). This commentary arrives at a similar conclusion: PMID: 17372595. We also agree with this statement and in future studies using larger animal models or higher dosage, examining the immunological profile should be strongly considered but are beyond the scope of this study.

6, Authors need to perform experiments to detect potential host immune response, such as IL2, IL15 or AAV antibody to confirmed the results in Line 327.

Please see response to the previous comment describing our reasoning for not performing an extensive analysis of the host immune response.

7, The off-target effect of ABE system is a topic of concern, and WGS was greatly influenced by the genetic background of the individual animal. Sequence analysis of the candidate off-target amplicons was needed to reveal no notable editing at any of the top 8-15 tested off-target sites.

We appreciate the reviewers great suggestion and have performed the suggested experiment. We took 7 candidate off-target loci based on in silico predictions taking into account regions with similarity to the gRNA sequence, as well as 9 other loci based on putative A/T > G/C edits detected in our WGS analysis. The main result is that we did not observe any differences between PBS and base editor-treated mice. These data reveal no off-target editing at these sites. For the 7 in silico predicted loci and 2 WGS loci, we did not observe any editing at all. 4 WGS loci turned out to be germline variants with >95% editing across all 10 mice. While we performed several filter steps (details in methods) to remove germline variants from the analysis, it is not surprising that some variants were not removed in the analysis. Due to the danger of removing potential true off-target edits, we refrained from increasing the threshold to remove more germline variants. For 3 WGS, we detected 1-15% of editing, however no increase in base editor treated mice compared to PBS was observed. We added text in the results and a new figure (**Sup. Fig. 6f**).

L409-16: “...We further analyzed 16 selected sites by amplicon-seq: 7 loci with highest sequence similarity to the gRNA used and 9 candidate A/T>G/C variants determined by WGS with the highest sequencing coverage (Sup. Fig. 6f). No SNVs were detected in the in silico predicted off-target sites. In addition, 4 out of 9 candidate loci from WGS were >90% mutated in both PBS and AAVMYO-ABE-treated mice and therefore are likely germline variants. For the remaining 5 SNVs, no difference in the percentage of editing was observed after AAVMYO-ABE injection compared to PBS. Overall, this data does not indicate the presence of ABE-induced DNA off-target edits...”

To further expand the off-target analysis, we have additionally performed an unbiased variant detection using bulk RNA-seq which enables the detection of transient (but potentially detrimental) RNA editing, a known concern of base editors. With this analysis, we uncovered a small but significant increase in A>G edits in R636Q mice treated with 8e-NRCH which could indicate rogue mRNA off-target editing by this base editor. However, similar to the WGS analysis, we did not find evidence for gRNA-dependent off-target editing. In future studies, the effect of systemic rogue off-target mRNA editing should be considered e.g. by dissecting the effect on coding sequence and mRNA stability.

L417-29: "...Since RNA editing has been reported as byproducts of ABEs^{42,43}, we also analyzed bulk RNA-seq data obtained 12 weeks after AAVMYO-ABE treatment of P635L and R636Q HOM mice. We confirmed high expression of the base editor in the heart and its absence in the liver (Sup. Fig. 7a) leading to high on-target and minor bystander editing for 8e-NRCH in P635L HOM (Sup. Fig. 7b). Unbiased variant detection was performed on the RNA-Seq data (Sup. Fig. 7c; details in the method) and a small but significant increase (from 17% to 19%) in the fraction of A>G mutations was observed only in the 8e-NRCH compared to PBS-treated R636Q HOM mice (Sup. Fig. 7d). No significant differences were observed in the other AAVMYO-ABE treated samples. Similar to WGS, we could not detect major sequence homology between the region surrounding the variants and the gRNA (Sup. Fig. 7e) indicating that the increased frequency of A>G mutations is not due gRNA-dependent effects. In summary, while our analysis prohibits the detection of random SNVs arising in a subset of cells, we do not find evidence that the base editing strategy induces systemic off-target editing..."

New Sup. Figure 7: RNA on and off-target editing. **a)** Base editor expression in heart and liver tissue 12 weeks after AAVMYO-ABE or PBS treatment in P635L or P636Q HOM mice. RPKM = reads per kilobase million. **b)** On-target edit (blue arrow) and bystander edit (red arrow) within the gRNA region targeting *Rbm20* on chromosome 19. Shown is the percentage of each base aligned to the positions plotted on the x-axis for R636Q and P635L HOM mice. In each row, three replicates were summed before calculating the fractions. **c)** Number of heart (H), liver (L) and common variants after filtering as described in the method section. **d)** Averaged relative amount of distinct types of SNVs identified as heart-specific. Significance tests were performed with a logistic regression model testing the difference in A<G vs. non-A<G ratios between pairs of ABE and PBS-treated samples. Error bars depict standard deviation. **e)** Number of mismatches to the gRNA and PAM sequence in the area of ± 30 bases around the variant start site. Variants from three replicates were summed up. Variants on the X or Y chromosome were excluded.

Minor points

1, Line73-74 “3% of patients with aggressive, early onset DCM” should provide reference.

We have added a reference (PMID: 22004663) to back up this statement.

2, Line98-99 Did mice carried both P635L and R636Q mutations? Sentences needed to be more accurate.

We modified this sentence to make it clear that two separate mouse lines, each with a different mutation, were created.

L98: "...we established two *Rbm20* knock-in mouse models ..."

3, The position of bystander need to be clearly in Sup. Fig. 3b etc.

We have added a cartoon to show the location of on-target and observed bystander edits in Fig. 2g, Sup, Fig. 3b, c and Sup. Fig. 4b.

4, The variant of number of active genes, total transcript counts and percentage of mitochondrial gene counts per cell were large (Sup fig5.a-c). Please provide the sequencing quality results and sample numbers in the supplementary.

We added a table depicting number of cells before and after QC as well as mapping statistics in Sup. Table 5.

5, AAVMYO exhibited superior efficiency and specificity in the musculature and liver detargeting. It was amazing and interested. Although previous work (PMID: 33116134) have been reported, author also needed to provide more detailed information about AAVMYO sequence or mutants in supplementary.

As noted by the reviewer, AAVMYO including the P1 peptide sequence (RGDLGLS) has been reported in PMID: 33116134. Furthermore, we refer the reviewer to PMID: 32105604 which contains further details on the design of this vector including peptide insertion site and flanking amino acids (the capsid was called AAV9P1 in this publication). Additional details are provided in PMID:35398994 including the primary sequence in Fig. 3 and a structural model in Supp. Fig. S2.

6, The vertical axis in the figure were confusing and did not told whether editing efficiency was at the DNA or RNA level, such as fig2. g, h and i. It should be shown clearly.

We have clarified whether editing was measured on DNA or RNA by adding this information in the axis text.

7, The figure legend should be clear about the animals numbers and tissues.

We clarified the number of animals and tissue in the figure legends.

Reviewer #2 (Remarks to the Author):

In this study, Grosch, et al generated Rbm20 knock-in mice resembling human RBM20 cardiomyopathy and established base-editing therapy using their muscle-specific AAVMYO vector combined with a variety of base-editing constructs. The authors demonstrated that base-editing therapy repaired genomic variant both in vitro and in vivo with high efficiency, corrected transcriptional profiles and abnormal cytoplasmic localization of Rbm20 protein, and finally improved physiological phenotypes. Furthermore, the authors proved that aberrant off-target editing was not observed using whole-genome sequencing. This study presents comprehensive experimental data based on step-by-step robust pilot experiments and provides a proof of concept of base-editing therapy for future clinical application. I have some minor comments.

We are grateful for the reviewers appreciation of our manuscript and have addressed the reviewer's points below.

1. In Sup. Figure 1g, mRNA expression levels of Col1a2 and Mmp2 are more than 2-fold increased from WT, although significant difference was not labeled. Is this correct?

We thank the reviewer for pointing out the missing statistical test for this data. Indeed, the upregulation of *Col1a2* and *Mmp2* in R636Q HOM mice is significant. We updated Sup. Fig. 1g and modified the text in the result section.

L139-41: "...Both histological analysis and gene expression did not uncover major signs of fibrosis in 16-week-old mutant mice except upregulation of the fibrosis marker *Col1a2* and *Mmp2* in R636Q HOM mice (Sup. Fig. 1g-i)..."

Modified Sup. Fig. 1g: Expression fold change compared to WT of [...] or fibrosis marker genes (g) determined by qPCR. N = 5-6 mice per genotype. Significant changes indicated and analyzed by unpaired t-tests. * p < 0.05, ** p < 0.01.

2. The authors stated the increased LVID in knock-in mouse models (line 136), but there is no significant difference in LVID among genotypes in in Sup. Figure 1k.

We thank the reviewer for pointing out this mistake and changed the text accordingly (see below). We also would like to highlight that we show the echo data after 24-weeks because initially, we had acquired data for both mutations only at this time point. For the revision, we added new data in **Sup. Fig. 1m and n**, where we performed time course echocardiography for the *Rbm20* mutant mice. The data clearly depicts a trend of larger LVID and cardiac volume across all time points and mouse mutant stains, compared to wild-type.

L142-46: "...We performed narcosis echocardiography, which confirmed that both mouse models exhibit a DCM phenotype with significantly reduced ejection fraction (**Fig. 1j**). However, they displayed only minor increase in cardiac volume (except in P635L HOM) and no significant change in the left ventricular internal diameter (LVID) (**Sup. Fig. 1k, l**)..."

3. Time dependent increase in the editing efficiency both in genomic DNA and cDNA shown in *Sup. Figure 3h* and *Figure 2i* (the efficiency of *in vivo* base editing was approximately 2-fold increased at 12 weeks compared to 6 weeks after injection) is interesting. Please explain the putative underlying mechanism in discussion.

We thank the reviewer for highlighting this indeed interesting observation. We had refrained from discussing it in detail since we observe this phenomenon only for the base editor SpRY and not for NRCH (**Sup. Fig. 3h**). Notably NRCH was conjugated to the hyperactive base editor domain Abe8e whereas SpRY was associated with the older version Abemax, which was available at the start of this study. We speculate that Abemax-SpRY takes longer to install the correct edit, meaning it could be expressed in the cell for some time before performing the gene repair whereas 8e-NRCH is more efficient. This would also explain the higher variability associated with Abemax-SpRY compared to 8e-NRCH. An alternative explanation could be an increase in the amount of virus reaching cardiomyocytes over the course of many weeks although we deem this unlikely and does not explain why we don't see differences in 8e- NRCH.

4. The highest levels of *in vivo* editing efficiency were achieved by high cell tropism of AAVMYO that only infects cardiomyocytes, as authors stated (line 231). Cell-type specific expression profiling of the viral transcripts derived from the AAVMYO-specific components using the authors' snRNA-seq data may further strengthen the beneficial effects of AAVMYO-based therapy.

This is a great suggestion and we have added an additional Supplementary figure (**Sup. Fig 5e, f**) showing the expression of the base editor constructs. Albeit overall lowly expressed, we clearly observe expression almost exclusively in cardiomyocytes. This observation confirms the myogenic tropism of AAVMYO and explains the high editing of RBM20 mRNA which is predominately expressed in cardiomyocytes. We added this data to the manuscript.

L356-8: "...In our snRNA-Seq data, we also analyzed the expression of the base editor complex itself and confirmed predominant targeting of cardiomyocytes by AAVMYO (**Sup. Fig. 5e, f**)..."

New Sup. Figure 5e, f: UMAPs (e) and quantification of the fraction of cells expressing the base editor construct (f) delivered by AAVMYO. N- and C-terminal base editor expression values were summed up. Values are either 0 (not expressed, grey) or 1 (expressed, red).

Reviewer #3 (Remarks to the Author):

The manuscript by Grosch et al examines the ability of an adenoviral gene transfer vector to target CRISPR agents in a tissue-specific manner to the heart/skeletal musculature. Previous investigations by the authors and others established that mutations in Rbm20, a factor that regulates splice-site choice during alternative splicing, can lead to a dilated cardiomyopathy (DCM) phenotype. Transgenic mice were developed encoding the P635L and R636Q mutations which reflect RBM20 mutations in DCM patients. In addition to the production of aberrantly spliced transcripts, cytoplasmic granules containing RBM20 were also found in the cardiomyocytes of these mice. Using knock-in mice of the P635L and R636Q RBM20 gene, the authors utilized an adeno-associated viral vector which specifically targets striated muscle to deliver and express CRISPR reagents that edited the RBM20 mutations to wild-type sequences. These edited sequences restored cardiac function and the transcriptional profile of wild-type hearts and cardiomyocytes. Delivering tissue-specific reagents for gene editing in the whole animal has been problematic in the past. The methods employed by these investigators address this problem and are logical extensions of their previous work in furthering gene editing processes for the correction and treatment of mutations associated with DCM, specifically the splicing factor gene RBM20. Overall, this is a very well-conceived and scientifically-sound investigation that integrates newly developing gene-editing techniques with potential for treatment of cardiovascular diseases. The research and methodology presented in this manuscript provides a comprehensive example of cardiomyocyte specific gene targeting to correct DCM mutations and analyses of the molecular, physiological, and cytological effects in isolated cardiomyocytes, iPSCs, and in vivo mice. The experiments are well-designed, with proper controls and clearly presented results. The discussion of the work is well integrated into the current state-of-knowledge in this field.

There are, however, several minor suggestions that would strengthen the manuscript.

We are thankful for the reviewer's kind evaluation of our work.

1. Lines 97 – 99: What is the amino acid identity between human and mouse RBM20 sequences? Also, a discussion of the changes on the protein structure and function that occur with the P635L and R636Q mutations (and orthologous changes in DCM patients) may enhance the understanding of why and how these mutations affect the alternative-splicing process in various genes.

Based on sequence alignments, there is a 77.2% identity between human and mouse RBM20 protein sequences (see graphic below). Of note, the RS-domain, which is also the mutation hotspot, is well preserved between these species. We have modelled and looked for potential structural changes in RBM20 harboring the P635L or R636Q mutations using Alphafold. However, no overt changes were visible. The mutations in the RS domain are believed to disrupt localization of RBM20. When mutant RBM20 localizes to the cytoplasm, and then cannot carry out splicing in the nucleus. In another study in review at *Nat. Communications*, we dissect the mechanism of mis-localization (and consequently mis-splicing) further and found a regulator of RBM20 import which exhibits position-dependent binding changes upon mutation in the RS-domain. In this paper, we show that different RBM20 mutations abrogate binding to this import protein at different levels, which is likely the underlying reason for differences in the disease phenotype between P633L (or P635L in mouse) and R634Q (or R636Q in mouse) mutations.

77.2% identity in 1231 residues overlap; Score: 4814.0; Gap frequency: 2.9%

human	1	MVLAAMSQADADPSGPEQDRVACSVPGARASPAAPSGPRGMQPPPPPPPPQAGLPQ	human	597	LQKKPGKAVAAIIQDIHSQERDMFREADRYGPERP
murine	1	MVLAVAMSQADADPSGPEQDRDCAVMPGVQGPSVQGOQMQPLPPP—PPQPQASLPQ	murine	599	LQKKPGKAVAAIIQDIHSQERDMFREADRYGPERP
		****			*****
		*****			*****
		*****			*****
human	61	IIQNAAKLLDKNPFVSNPNPLPSPASLQLAQLLHRLKLAQTAVTNNTAAATVL	human	717	QLDKAELDERPEGGPRHREKYPKSGSPNLSVSSYSKREDGYRKEPKAKSDKYLKQQ
murine	59	IIQNAAKLLDKSPFVSNPNPLPSPASVQLAQIQAQLLHRLKMAQTAVTNNTAAATVL	murine	703	QLDKAELDERLEGGRYREKYLKSGSPGLHSVSGYKREDGYRKEPKAKLDKYPKQQ
		*****			*****
		*****			*****
		*****			*****
human	121	NQVLSKVAMSQPLFNQLRHPSVITGPHGHAGVPQHAHAIPSTRFSPNAIAFSPSPQTRGP	human	777	DAPGRSRKDEARLRERHPHDDSGKEDLGPVKVTRAPGAKAKQNEKNTKTRDRDQE
murine	119	NQVLSKVAMSQPLFNQLRHPSVLTGTAHGPTGVSQHAASVPSAHFPAIAFSPSPQTRGP	murine	763	DVPGRSRKEEARLRERHPHDDSGKEDLGPVKVTRAPGAKAKQNEKNTKTRDRDQE
		*****			*****
		*****			*****
		*****			*****
human	181	GPSMNLNQPPSAMVHPFTGVMPTPGQPAVILGIGKTGPAPATAGFYEGKASSGQTY	human	837	GADDRKENTMAENEAGKEEQEGMEESPQSVGRQKEAEFSDPENTRTKKEQDWESEAE
murine	179	GPSVSLPSQPPNMMVHTFSGVMPQTPAQPAVILSLGKAGTPATTGFYDYGKANSQGY	murine	823	GADDKESQLAENEAGAEQEGM-----VGIQEGTESCDPENTRTKKGQCDGSGEPE
		*****			*****
		*****			*****
		*****			*****
human	241	GPETDQGPGLPSSAST—SGSVTYEGHYSHTGQDQAAAFSKDFGPNSSQSHVAGFPA	human	897	GESWYPTNMEELVTVEVEEEDFIVEPDIPELEEIVIDQDKKICPETCLCVTTLDLD
murine	239	GSETEGQGPGLPASASATASGSMTYEGHYSHTGQDQPAFSDFYGPNAAQPHIAGGFPA	murine	877	GDNWYPTNMEELVTVEVEEEDFIMEPDLPELEEIVIDQDKKILPKICTCVATLGLD
		*****			*****
		*****			*****
		*****			*****
human	299	EQAGGLKSEVGLLQGTNSQWESPHGFSGQSKPDLTAGP—MWPPIHNPYELDPEEPTS	human	957	LAQDFPKEGKAVGNGAAEISLKSRELPSASTSCPSMDVEMPGLNLDARKEPAESETG
murine	299	DQTSKMGDVGGLLQGTNSQWESPHGFSGQSKPDLTAGP—MWPPIHNPYELDPEEPTS	murine	936	LAKDFTKQG—ETLNGDAELSLKLPQVPSASCPNDTLEMPGLNLDARKEPAESETG
		*****			*****
		*****			*****
		*****			*****
human	358	DRTPPSFGRLNNSKQGFAGARRAKEDQALLSVRPLQAHLENDFHGVAPLHLPICISIC	human	1017	LSLESDCYEKEAGVSSDVHPAPTVMQSSPKAEERARQSPFVDDCKTRGTPEDGA
murine	359	DRAPPAFGRSLNNSKQGFAGARRAKEDQALLSVRPLQAHLENDFHGVAPLHLPICISIC	murine	995	LSLEVSNCYEKEAGVSSDVSLAPAVQMQSSPKAEERARQSPFVDDCKTRGTPEDGA
		*****			*****
		*****			*****
		*****			*****
human	418	DKKVFDLKDWELHVKGLHAQKCLVFSENAGIRCILGSAEGLTLCASPSTAVYNPAGNE	human	1077	CEGSPLEEKASPIETDLQNAQCEVLTPENSRYEMKSLVRSPEYTEVELKQPLSLPS
murine	419	DKKVFDLKDWELHVKGLHAQKCLVFSENAGIRCILGSAEGLTLCASPSTAVYNPAGNE	murine	1055	HEASPLEGKASPTESDLQSQACRE-----NPRYMEVKSINVRSPFTEAELEKPLSLPS
		*****			*****
		*****			*****
		*****			*****
human	478	YASNLGTSYVPIPARSFTQSSPTFPLASVGTTF—QRKGAGRVVHICNLPEGSCTENDVI	human	1137	WEPEDEVFSELSIPLGVEFVVPRTGFYCKLCGLFYTSEETAKMHSRCSAVHYRNLQKYL
murine	479	YTSNLGTSYAAIPTRAFQSNPVFSPASSGTSFAAQKRGAGRVVHICNLPEGSCTENDVI	murine	1110	WEPE—VFSELSIPLGVEFVVPRTGFYCKLCGLFYTSEEAHVSHCRSTVHYRNLQKYL
		*****			*****
		*****			*****
		*****			*****
human	537	NLGLPFGKVTNYILMKSTNQAFLEMAYTEAAQAMVQYQKSAVINGEKLLIRMSTRYKE	human	1197	LAEEGLKETEGADSPRDESGIVPRFERKRL
murine	539	NLGLPFGKVTNYILMKSTNQAFLEMAYTEAAQAMVQYQKSAVINGEKLLIRMSTRYKE	murine	1169	LAEEGLKETEGADSPRDESGIVPRFERKRL
		*****			*****
		*****			*****
		*****			*****

Sequence comparison (performed with *Expsy*) between murine and human *Rbm20* protein sequence. Figure not included in the manuscript.

2. Lines 118 – 119: There should be additional discussion of why there was little overlap between P635L and R36Q DEGs and DSGs.

One possibility of the low overlap is that there are mutation-specific perturbations of gene expression. In addition, biological variation between mice could add to the fact that we observe gene expression changes that are not directly related to the *Rbm20* mutation. Since each mouse was sequenced on average with more than 100 Mio. reads, smaller changes in gene expression are observed and measured as significant. This is supported by the fact that the non-overlapping genes have a significantly higher p-value compared to overlapping genes and therefore are more likely to be noise. We have added this analysis as new supplementary figure in the manuscript (**Sup. Fig. 1f**). Moreover, we changed the result section to discuss this and are careful to avoid overstating our conclusions.

L119-25: "...Notably, approximately half of all DEGs and DSGs did not overlap between P635L and R636Q HOM (**Sup. Fig. 1e**). While these specific genes could suggest the presence of mutation-specific downstream processes, their p-values were higher on average than for overlapping genes (**Sup. Fig. 1f**). This is consistent with the detection of subtle changes in transcript abundance arising due to biological variation, such as between mice, or other confounding factors were detected by our deep RNA-seq with 100 Mio. reads on average per genotype...."

New Sup. Fig. 1f:) P-values (cut-off $p_{\text{adjust}} < 0.05$) of DEGs unique or common between P635L (red) and R636Q HOM (grey). Significant changes analyzed by unpaired t-tests. **** $p < 0.0001$.

3. Lines 120 – 122: In Fig. 1g, some genes (i.e. Ttn) are listed more than once – why?

The plot in Fig. 1g depicts splice events and for some genes (e.g. TTN) multiple splice events can be found to be mis-regulated upon *Rbm20* mutation. We decided to show all events, irrespective if they arise from the same gene. We clarified that these represent splice isoforms in the figure legend.

L168-9: “...Multiple splice events per gene are depicted if they match the selection cut-off...”

4. Line 136 “increased cardiac... (LVID)”: In Supp. Fig 1j, k – there appears to only be a significant change in cardiac volume in the P635L Homo samples, not in the other genotypes (Supp. Fig 1j). As shown, there are no changes in the LVID for any genotypes in Supp. Fig 1k. The text in the manuscript should be changed to reflect this.

We thank the reviewer for pointing out this mistake and changed the text accordingly (see below). We also would like to highlight that we show the echo data after 24-weeks because initially, we had acquired data for both mutations only at this time point. For the revision, we added new data in **Sup. Fig. 1m and n**, where we performed time course echocardiography for the *Rbm20* mutant mice. The data clearly depicts a trend of larger LVID and cardiac volume across all time points and mouse mutant strains compared to wild-type. We modified the text accordingly.

L142-46: “...We performed narcosis echocardiography, which confirmed that both mouse models exhibit a DCM phenotype with significantly reduced ejection fraction (**Fig. 1j**). However, they displayed only minor increase in cardiac volume (except in P635L HOM) and no significant change in the left ventricular internal diameter (LVID) (**Sup. Fig. 1k, l**)...”

5. Lines 186 - 187: the wording of “partially approached” is awkward – this should be change. Also, in Fig 2e, the error bar for the WTd32 sample is very large – can this be tightened up to ascertain whether there was statistical significance among the samples?

We changed the wording to reflect more accurately the trend of the spliced and unspliced isoforms after base editing. Indeed the variation is quite big which likely is due to the fact that the genes are relatively low expressed (relative to GAPDH) and therefore small changes in expression cause bigger variation. This is the reason, we performed the analysis at two separate time points, which showed a similar trend in upregulation of spliced and downregulation of unspliced isoforms between mutant iPSC-CMs and after gene repair. We updated the graph in Fig. 2e by highlighting the statistical significance (if present) as the reviewer suggested.

L197-200: “...After differentiation to iPSC-CMs, the expression levels of spliced isoforms of *TTN* and *IMMT*, prominent RNA targets of *RBM2035*, were upregulated whereas the unspliced isoforms were downregulated in base edited cells suggesting that base editing restored *RBM20*-related splice defects (Fig. 2e)...”

Modified Fig. 2e, f: Expression of spliced and unspliced isoforms of *TTN* and *IMMT* in parental, R634Q and edited R634Q iPSC-CMs differentiated for 15 and 32 days. N=3-4 independent differentiations. Significant changes compared to R634Q indicated when present and analyzed by unpaired t-tests. * p< 0.05, ** p< 0.01, **** p< 0.0001.

6. Line 196: Please provide the rationale for selecting the P635L HOM mice for further testing and analyses. The P636Q mice displayed a more severe phenotype. Why not select both Het and HOM mice of both genotypes?

We focussed on homozygous mice since they show a more pronounced phenotype compared to the heterozygous animals, clarified in the main text. We could not detect a significance difference in phenotype between P635L and R636Q HOM and therefore choose just one mouse line, namely P635L HOM, for initial optimizations of the AAVMYO-base editor delivery in Fig. 2. Subsequently, both mutant lines were chosen for AAVMYO treatment and phenotypic characterization.

L153-5: “...For subsequent rescue strategies, we focused on P635L and R636Q HOM mice since they showed a more pronounced molecular and physiological defect enabling better quantification of the efficacy of the base editor treatment...”

Reviewers' Comments:

Reviewer #1:

Remarks to the Author:

I really like your work.

Reviewer #2:

Remarks to the Author:

In the revised manuscript, the authors appropriately addressed the comments. Specifically, the additional transcriptional profiling of the base editor complex itself using snRNA-seq data clearly demonstrated the predominant expression of AAVMYO in cardiomyocytes and strengthened the study.

Reviewer #3:

Remarks to the Author:

The authors have comprehensively addressed all of the concerns of this reviewer. With these changes and addition of new data/figures, the manuscript has been improved significantly. There are no additional concerns of this reviewer.